# Formin-2 drives polymerisation of actin filaments enabling segregation of apicoplasts and cytokinesis in *Plasmodium falciparum*

Johannes Felix Stortz[1], Mario Del Rosario[1], Mirko Singer[2], Jonathan M Wilkes[1], Markus Meissner[1,2]*, Sujaan Das[1,2]*

[1]Wellcome Centre for Integrative Parasitology, Institute of Infection, Immunity & Inflammation, University of Glasgow, Glasgow, United Kingdom; [2]Faculty of Veterinary Medicine, Experimental Parasitology, Ludwig Maximilian University, Munich, Germany

*For correspondence:
Markus.Meissner@para.vetmed.
uni-muenchen.de (MM);
Sujaan.Das@para.vetmed.uni-
muenchen.de (SD)

**Competing interests:** The authors declare that no competing interests exist.

**Abstract** In addition to its role in erythrocyte invasion, *Plasmodium falciparum* actin is implicated in endocytosis, cytokinesis and inheritance of the chloroplast-like organelle called the apicoplast. Previously, the inability to visualise filamentous actin (F-actin) dynamics had restricted the characterisation of both F-actin and actin regulatory proteins, a limitation we recently overcame for *Toxoplasma* (Periz et al, 2017). Here, we have expressed and validated actin-binding chromobodies as F-actin-sensors in *Plasmodium falciparum* and characterised *in-vivo* actin dynamics. F-actin could be chemically modulated, and genetically disrupted upon conditionally deleting *actin-1*. In a comparative approach, we demonstrate that Formin-2, a predicted nucleator of F-actin, is responsible for apicoplast inheritance in both *Plasmodium* and *Toxoplasma*, and additionally mediates efficient cytokinesis in *Plasmodium*. Finally, time-averaged local intensity measurements of F-actin in *Toxoplasma* conditional mutants revealed molecular determinants of spatiotemporally regulated F-actin flow. Together, our data indicate that Formin-2 is the primary F-actin nucleator during apicomplexan intracellular growth, mediating multiple essential functions.
DOI: https://doi.org/10.7554/eLife.49030.001

## Introduction

The phylum Apicomplexa includes a variety of obligate intracellular parasites, which invade into and replicate inside mammalian cells, causing immense disease burden in humans and in commercially important livestock. One of its notorious members, the malaria parasite *Plasmodium falciparum*, is a major health concern in developing nations, causing ~500,000 deaths annually (*White et al., 2014*). Another member, *Toxoplasma gondii* is a highly successful parasite infecting almost a third of the global human population and can be fatal in immunocompromised patients (*Torgerson and Mastroiacovo, 2013*).

Actin is one of the most abundant proteins in eukaryotic cells. Due to its ability to form polymers, this cytoskeletal protein is involved in numerous processes such as cell motility, cytokinesis, organellar and vesicular transport, secretion and endocytosis (*Svitkina, 2018*). Actins encoded by apicomplexan parasites are highly divergent compared to canonical actins from other eukaryotes (*Douglas et al., 2018*). *In-vitro*, apicomplexan actins form only short, unstable polymers due to different polymerisation kinetics, caused by variation of certain key amino acids otherwise conserved in metazoans (*Kumpula and Kursula, 2015*). However, until recently, an analysis of filamentous actin (F-actin) localisation and dynamics in apicomplexan parasites was hindered by the unavailability of

F-actin sensors (*Tardieux, 2017*), a limitation recently overcome by the expression of F-actin binding chromobodies in *T. gondii* (*Periz et al., 2017*). Intriguingly, in this parasite, F-actin can form an extensive intra-vacuolar network that appears to be involved in material exchange and synchronisation of parasite division (*Periz et al., 2017*).

Until recently, studies on apicomplexan F-actin focused on its critical role during host cell invasion and gliding motility (*Soldati et al., 2004*; *Baum et al., 2008a*), where it is believed to provide the force for both processes (*Frénal et al., 2017*). However, recent studies using conditional mutants for *actin-1* in two apicomplexans, *P. falciparum* and *T. gondii* highlight additional critical roles of F-actin during intracellular parasite development (*Das et al., 2017*; *Periz et al., 2017*; *Whitelaw et al., 2017*). Intriguingly, some functions, such as inheritance of the chloroplast-like organelle, the apicoplast, appear to be conserved (*Andenmatten et al., 2013*; *Egarter et al., 2014*; *Das et al., 2017*; *Whitelaw et al., 2017*), while differences for the dependency of F-actin can be observed for other critical steps of the asexual life cycle. For example, host cell invasion is possible without *actin-1* (albeit at highly reduced levels) in case of *T. gondii* (*Andenmatten et al., 2013*; *Egarter et al., 2014*; *Whitelaw et al., 2017*), while it is completely blocked in case of *P. falciparum* (*Das et al., 2017*). In contrast, *P. falciparum* does not require actin dynamics for egress from the host cell (*Das et al., 2017*; *Perrin et al., 2018*), while it is essential for *T. gondii*. Additionally, completion of cytokinesis in *P. falciparum* is dependent on actin dynamics (*Das et al., 2017*), while no such dependency has been noted for *T. gondii*. The functions of *P. falciparum* actin-1 have been summarised in a table in *Figure 1A*.

Of the two actin genes present in *P. falciparum* (*Gardner et al., 2002*), only *actin-1* (*pfact1*, GeneID: PF3D7_1246200) is expressed in all life-cycle stages and is the only actin expressed during asexual replicative stages, whereas *actin-2* (GeneID: PF3D7_1412500) expression is confined to the sexual gametocyte and insect stages (*Deligianni et al., 2011*; *Vahokoski et al., 2014*). *P. falciparum* undergoes a 48 hr asexual replicative cycle in the intermediate human host where it invades into, grows and replicates within erythrocytes, causing all clinical manifestations of the disease. After invasion, the merozoite form of the parasite (similar to *T. gondii* tachyzoites) establishes itself within a parasitophorous vacuole (PV), loses its ovoid shape to become amoeboid and feeds on host haemoglobin creating a food vacuole (FV) where haem is detoxified (*Grüring et al., 2011*). Interestingly, actin has been implicated in transport/fusion of haemoglobin-filled vesicles (*Elliott et al., 2008*; *Smythe et al., 2008*). The parasite then replicates by a process best described as internal budding, where daughter parasites develop within the mother (*Francia and Striepen, 2014*). In the case of *T. gondii*, only two daughters are formed at a time in a process called endodyogeny. In contrast, malaria parasite replication within the erythrocyte, termed schizogony, results in the formation of 16–32 merozoites at once. Towards the end of a replicative cycle the parasite forms its invasion-related organelles de novo: the inner membrane complex (IMC), micronemes and rhoptries. In contrast, parasite mitochondria and the apicoplast undergo growth and division and are trafficked into each daughter cell (*Bannister et al., 2000*). Although endodyogeny and schizogony appear different, it is believed that both processes use a conserved molecular machinery. Indeed, independent studies identified the same factors to be critical for both replicative modes (*Francia and Striepen, 2014*).

Despite this, it could be assumed that differences, especially with respect to vesicular transport processes such as endocytosis and intra-vacuolar parasite communication, exist to adapt to different replication modes. This puts F-actin in the spotlight as it plays a central role in these processes in other eukaryotes (*Svitkina, 2018*). We recently characterised a conditional mutant of PfACT1 and observed that in good agreement with the function of actin in *T. gondii* (*Andenmatten et al., 2013*; *Jacot et al., 2013*), inheritance of the apicoplast is compromised during schizogony (*Das et al., 2017*). While the phenotypic analysis of conditional mutants is useful to identify conserved and unique functions of F-actin in apicomplexans, the inability to visualise F-actin in these parasites led to models, sometimes conflicting with each other and with the canonical behaviour of F-actin in other eukaryotes.

Common actin-labelling probes such as Phalloidin do not label apicomplexan actin and LifeAct could not be successfully expressed in these parasites (*Periz et al., 2017*). Recently, actin-binding

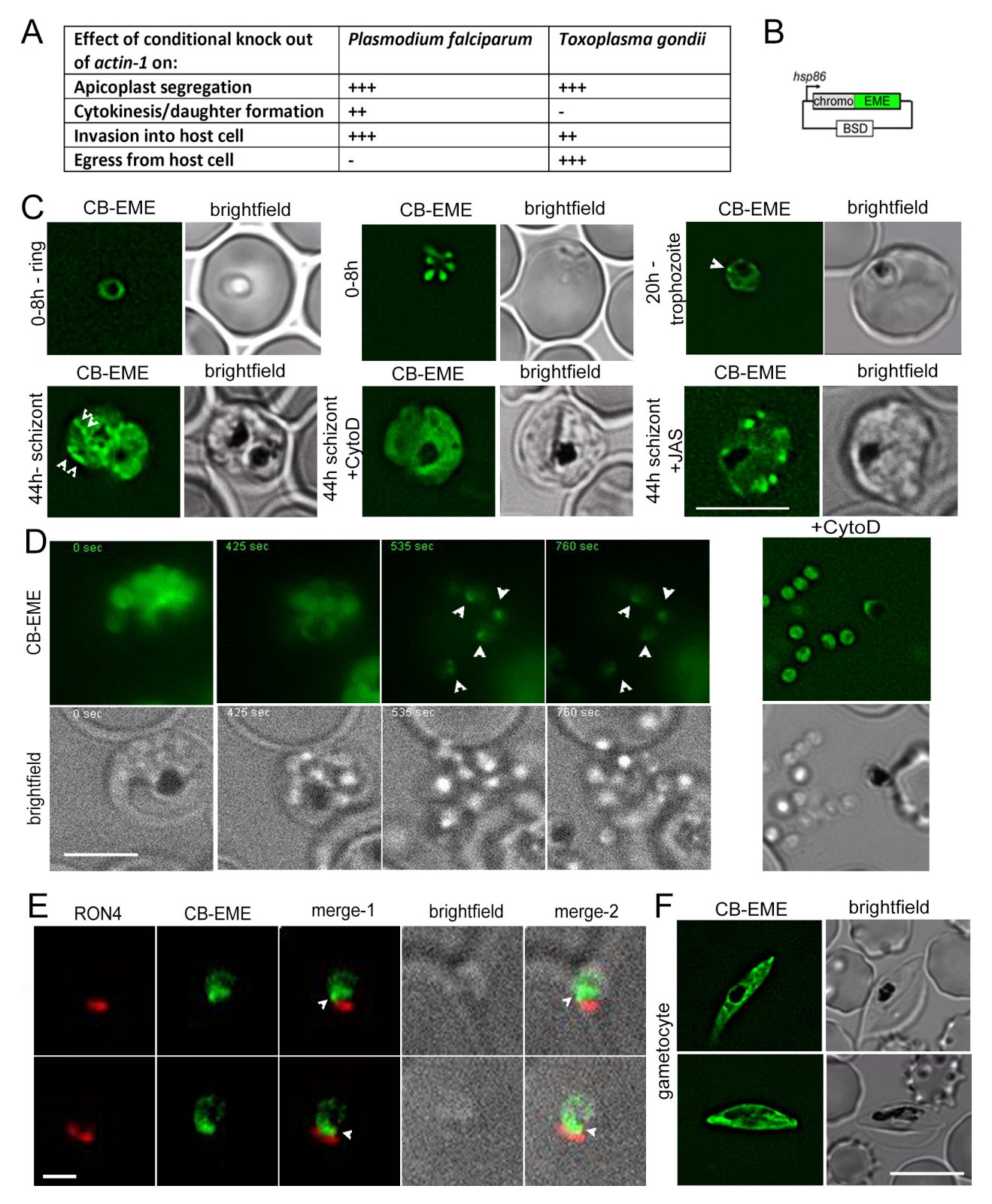

**Figure 1.** Chromobody-Emerald constructs label F-actin structures throughout the *P. falciparum* lifecycle. (**A**) Table summarising the functions of actin in *P. falciparum* and *T. gondii* (*Andenmatten et al., 2013*; *Das et al., 2017*; *Whitelaw et al., 2017*). Upon conditional disruption of *actin-1* in either organism, highly penetrant phenotypes observed are labelled as +++, moderate phenotypes as ++, and no effect on phenotype with -. (**B**) Chromobody construct used in this study under the *hsp86* promoter with a C-terminal emerald tag (CB-EME). See also *Figure 1—figure supplement 1*

*Figure 1 continued on next page*

*Figure 1 continued*

for a chromobody construct with a C-terminal halo tag (CB-HALO) and its characterisation. Blasticidin-S-deaminase (BSD) confers resistance to blasticidin. (C) CB-EME labels actin filaments throughout the 48 hr asexual life cycle with, in ring stages (0–8 hr), trophozoite stages (20 hr) and in 44 hr schizont stages. White arrows mark structures, in all likelihood F-actin, which disappear upon cytochalasin-D treatment (+CytoD), and islands of F-actin are stabilised upon jasplakinolide treatment (+JAS). Scale bar 5 µm. See also *Videos 1*, *2* and *3* and *Figure 1—figure supplement 2*. (D) Time lapse imaging of schizonts undergoing egress from the host cell, reveals a bright fluorescent signal (CB-EME) of F-actin (white arrows) at a polar end of the merozoite, appearing immediately after host cell rupture (occurs in 56 ± 9% egressed merozoites, N = 260 from three independent experiments). See also *Video 4*. Cytochalasin-D treatment completely prevents the polar polymerisation of F-actin in all cells (+CytoD). Scale bar 5 µm. (E) IFA of invading merozoites with the junction marker RON4 shows CB-EME staining close to the RON4 stain, implying that F-actin polymerises at the apical end prior to invasion. Scale bar 1 µm. (F) The F-actin network and dynamics can be visualised in gametocytes (see also *Video 5*). Brightfield images provided in greyscale alongside. Scale bar 5 µm.

DOI: https://doi.org/10.7554/eLife.49030.002

The following source data and figure supplements are available for figure 1:

**Source data 1.** Source data for table in *Figure 1A*.
DOI: https://doi.org/10.7554/eLife.49030.005
**Figure supplement 1.** CB-HALO labels the F-actin network similar to CB-EME.
DOI: https://doi.org/10.7554/eLife.49030.003
**Figure supplement 2.** Rapid actin dynamics are visible in 20 hr old trophozoites during intracellular growth.
DOI: https://doi.org/10.7554/eLife.49030.004

single-domain nanobodies tagged to fluorescent probes, called actin-chromobodies, were successfully expressed in *T. gondii* and shown to have minimal effect on actin dynamics (*Periz et al., 2017*), as also demonstrated in other eukaryotic cells (*Rocchetti et al., 2014*; *Panza et al., 2015*; *Melak et al., 2017*). Furthermore, these chromobodies could faithfully label purified *Plasmodium* actin in *in-vitro* experiments (*Bookwalter et al., 2017*).

Here we adapted the actin-chromobody technology to *P. falciparum* and demonstrate for the first time the localisation, dynamics and role of F-actin for parasite development in asexual stages. Interestingly, we find F-actin in proximity to the apicoplast throughout intracellular growth, leading to the question of which actin regulatory proteins are involved in this process. Most actin nucleation proteins such as the Arp2/3 complex and the WAVE/WASP complex, and actin cross-linkers such as α-actinin and fimbrin are missing in apicomplexans (*Baum et al., 2006*; *Schüler and Matuschewski, 2006*). Two conserved nucleators found in *P. falciparum* are the formins, Formin-1 (PfFRM1) and Formin-2 (PfFRM2) which localise to distinct compartments in the cell (*Baum et al., 2008b*). Orthologs of both formins have been implicated in host cell invasion in *T. gondii* (*Daher et al., 2010*), with *T. gondii* Formin-2 (TgFRM2) also being implicated in apicoplast maintenance (*Jacot et al., 2013*) – leading to inconsistencies in reports and questions whether the two formins have conserved or divergent functions in both parasites. Indeed, a recent study, using the F-actin chromobody as described previously (*Periz et al., 2017*), suggested distinct, non-overlapping functions for the three formins in *T. gondii* (*Tosetti et al., 2019*).

Here we performed a careful comparison of the role of Formin-2 in *P. falciparum* and *T. gondii* and demonstrate that it localises in close proximity to apicoplasts in both parasites. Conditional disruption of Formin-2 not only results in a complete abrogation of actin dynamics in *P. falciparum* and a loss of an intracellular F-actin polymerisation centre in *T. gondii*, it also leads to defects in apicoplast inheritance in both genera. In contrast to *Toxoplasma*, Formin-2 is additionally involved in completion of cytokinesis in *P. falciparum*. Together our study highlights conserved and distinct roles of Formin-2 in the intracellular development of apicomplexan parasites.

## Results

### Cellular expression of actin-chromobodies in *P. falciparum* enables the visualisation of an actin network throughout the asexual development of *P. falciparum* and in gametocytes

Actin-chromobodies label F-actin structures in *P. falciparum* asexual stages

Actin-chromobodies (Chromotek) were expressed under control of the *heat shock protein 86 (hsp86)* promoter (*Crabb and Cowman, 1996*) to obtain expression throughout the 48 hr asexual

life cycle. We succeeded in generating parasites stably expressing actin-chromobodies tagged either with the emerald tag (CB-EME) (*Figure 1B*), or the halo tag (CB-HALO) (*Figure 1—figure supplement 1*), indicating that the expression of these constructs does not have a major deleterious impact on the fitness of *P. falciparum* as previously reported for *Toxoplasma* (*Periz et al., 2017*) and other eukaryotes. The halo tag allowed visualisation of F-actin in live parasites by use of the red ligand Halo-TMR. Dynamic filamentous structures were evident in both CB-EME (*Figure 1*) and CB-HALO expressing parasites (*Figure 1—figure supplement 1*) throughout the 48 hr asexual life cycle (see also *Videos 1*, *2*, *3* and *4*) and in gametocytes (*Video 5*). These structures could be completely disrupted by treatment with F-actin destabilising drug cytochalasin-D (30 min, final concentration 1 μM) or stabilised by addition of the depolymerisation inhibitor Jasplakinolide (JAS) (30 min, final concentration 1 μM) (*Figure 1C* bottom panels and *Video 3*), demonstrating that chromobodies bind F-actin structures in *P. falciparum*. However, while both chromobody versions labelled similar structures, we found that expression of CB-EME resulted in a better signal-to-noise ratio. This is probably because no permeable, fluorescent ligand needs to be added for visualisation. Therefore, for the rest of this study, results for parasites expressing CB-EME have been presented. Some F-actin structures were highly dynamic, changing within a time-scales of seconds, while other structures appeared stable over tens of seconds (*Video 3*, see also quantifications below).

After erythrocyte invasion, the parasite immediately loses its ovoid zoite structure and becomes an amoeboid ring-stage parasite. These young parasites are highly dynamic and can switch between various shapes forming multi-lobed structures, possibly mediated by their cytoskeletal networks (*Grüring et al., 2011*). On observing chromobody-expressing parasites during ring and early trophozoite stages we noted F-actin rich islands at the periphery of ~60% parasites (n = 50) (*Figure 1—figure supplement 2A* and *Video 2*). These F-actin accumulations were highly dynamic, changing in order of seconds (*Figure 1—figure supplement 2A* and *Video 2*). Upon treatment with the F-actin depolymerising drug cytochalasin-D (30 min, final concentration of 1 μM) the peripheral, highly dynamic accumulations completely disappeared (*Figure 1—figure supplement 2B* lower panel), while the multi-lobed structures of the parasite were not disrupted (*Video 6*). The F-actin stabilising drug jasplakinolide also disrupted peripheral F-actin and resulted in formation of stable thick filaments (*Figure 1—figure supplement 2B* lower panel), implying the requirement of dynamic regulation of F-actin for peripheral F-actin accumulations. The physiological relevance of the observed structures is currently unclear.

## Apical polymerisation of F-actin in merozoites following egress

Next, we wished to analyse the fate of the observed F-actin network upon parasite egress. We synchronised parasites with a 2-step Percoll and sorbitol treatment and harvested schizonts at 44 hr post-invasion. Reversible inhibitors of protein kinase G, Compound-1 and −2, stall schizont development at very mature stages without allowing

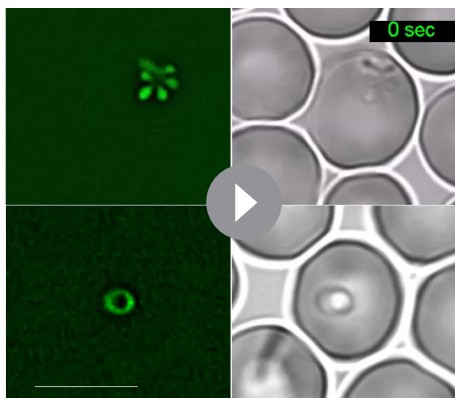

**Video 1.** Rapid shape changes of ring stages of *P. falciparum* expressing CB-EME (green). Acquisition time is shown in seconds. Scale bar 5 μm.
DOI: https://doi.org/10.7554/eLife.49030.006

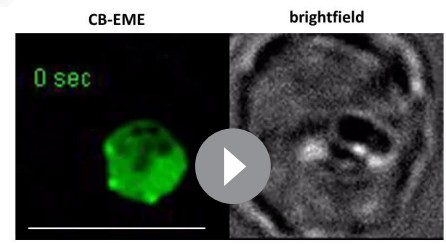

**Video 2.** Dynamic Filopodia-like F-actin extensions from the parasite edges into the RBC cytosol. Acquisition time is shown in seconds. Scale bar 5 μm.
DOI: https://doi.org/10.7554/eLife.49030.007

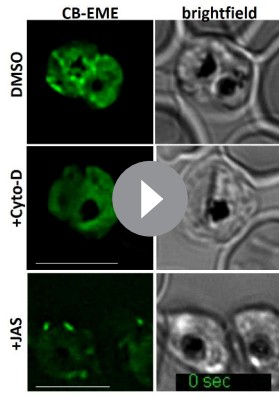

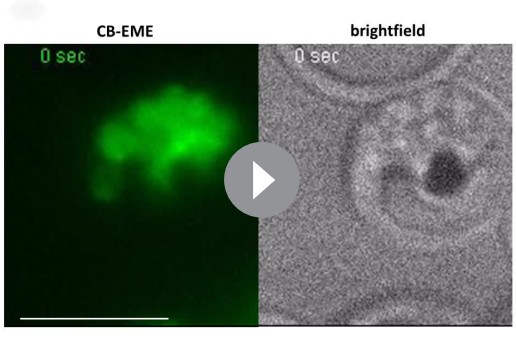

**Video 3.** Dynamic actin filaments in CB-EME expressing parasites (DMSO) are disrupted upon addition of cytochalasin-D and stabilised upon addition of jasplakinolide. The green channel shows CB-EME expression. Brightfield images also shown. Acquisition time is shown in seconds. Scale bar 5 µm.
DOI: https://doi.org/10.7554/eLife.49030.008

**Video 4.** Polar polymerisation of F-actin at the merozoite tip following egress. Time lapse images of a representative schizont which undergoes egress, followed by polymerisation of F-actin at the merozoite edge (white arrows appearing). Images (green channel, CBEME) and brightfield (greyscale) were acquired every 5 s. Acquisition time is shown in seconds. Scale bar 5 µm.
DOI: https://doi.org/10.7554/eLife.49030.009

them to undergo egress (*Collins et al., 2013b*). We treated highly mature schizonts with Compound-2 for 4 hr to allow them to fully mature without undergoing egress. Upon washing away Compound-2, the parasites egressed normally with the concomitant appearance of F-actin accumulation at the apical tip of the parasite (*Figure 1D*, *Video 4*). Cytochalasin-D treatment (30 min, final concentration 1 µM) allowed normal egress of parasites, as previously observed (*Weiss et al., 2015*), but completely abrogated F-actin polymerisation at the apical tip following egress (*Figure 1D*,+CytoD). Furthermore, we performed IFA on invading merozoites using rhoptry neck protein 4 (RON4) as a junctional marker. We verified that F-actin accumulates just behind the RON4 ring (*Figure 1E*) confirming previous observations made with an antibody preferentially recognising F-actin in *P. falciparum* (*Riglar et al., 2011*; *Angrisano et al., 2012*).

## F-actin in gametocytes

In contrast to asexual parasites, gametocytes express both PfACT1 and actin-2, and exhibit F-actin staining along the length of the parasite and at the tips (*Hliscs et al., 2015*). Upon expression of CB-EME, gametocytes showed intense dynamic F-actin structures at their tips and running along the whole body of the cell (*Figure 1F* and *Video 5*). Importantly, this dynamic network appears very similar to the one reported by *Hliscs et al. (2015)*, which has been shown to lie beneath the IMC of the gametocyte. It is important to note that actin-chromobodies do not distinguish between PfACT1 and actin-2 and therefore the observed filaments could potentially be composed of both proteins. We therefore confirm previous data obtained with antibodies directed against F-actin (*Hliscs et al., 2015*) and show that during the gametocyte stage, F-actin forms a dynamic and extensive network that passes through the whole cell and is enriched at the tips of the parasite.

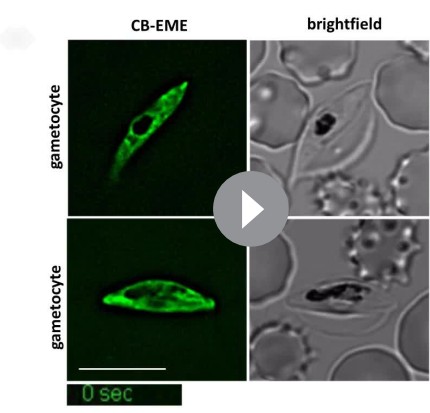

**Video 5.** F-actin dynamics in gametocytes. Two representative examples of gametocytes expressing CB-EME show dynamic filaments running along the parasite length and enriched at the tips. Acquisition time is shown in seconds. Scale bar 5 µm.
DOI: https://doi.org/10.7554/eLife.49030.012

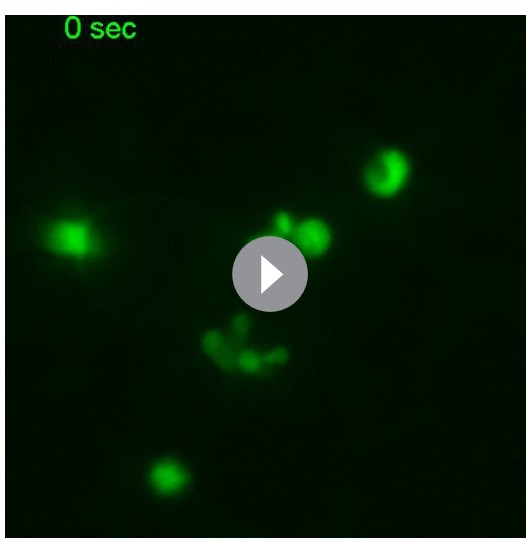

**Video 6.** Multilobular structures of trophozoites are not lost upon addition of cytochalasin-D.
DOI: https://doi.org/10.7554/eLife.49030.010

## Super-resolution microscopy reveals the spatial arrangement of F-actin

For better visualisation of F-actin structures, we used super-resolution microscopy (SR-SIM) which revealed a complex F-actin network in these parasites (*Figure 2A* and *Figure 2—figure supplement 1*). Interestingly, F-actin was prominent around the FV of the parasite (*Figure 2A* and *Figure 2—figure supplement 1*, see also *Figure 2—figure supplement 2* lower left panel), which is also the basal end of the newly-formed daughter parasites during the final stages of cytokinesis. When we co-stained the actin-chromobody-labelled network with an antibody raised against parasite actin (*Angrisano et al., 2012*), similar structures as seen with chromobody were apparent (*Figure 2A*). The staining was however not identical perhaps due to the masking of antibody binding sites by other binding proteins, which was also observed previously for *T. gondii* actin (*Periz et al., 2017*). Upon quantification of co-localisation of the CB-EME signal with that of the actin-antibody, we obtained a Pearson's R value of 0.6, in comparison to R values of 0.2 when the CBEME signal was tested for co-localisation with the DAPI signal. IMC markers GAP45 and MTIP showed normal staining in CB-EME expressing parasites (*Figure 2—figure supplement 1B*). Together, our data show that expression of chromobodies does not cause significant phenotypic effects and allow reliable labelling of the F-actin cytoskeleton.

## Chromobodies label authentic F-actin structures: CB-EME-labelled filaments disappear upon disruption of PfACT1

Although *P. falciparum* parasites possess two actin genes *pfact1* and *actin-2*, PfACT1 is the only protein expressed during the asexual life cycle (*Vahokoski et al., 2014*). In order to confirm that chromobodies label authentic F-actin structures built from polymerisation of PfACT1, we transfected the chromobody constructs pCB-EME and pCB-HALO (*Figure 1B* and *Figure 1—figure supplement 1A*) into a conditional mutant of PfACT1 (loxPPfACT1) (*Das et al., 2017*) (*Figure 2B*). Upon activation of DiCre with rapamycin (RAP), the *pfact1* locus is excised together with loss of PfACT1 protein within 35 hr (*Das et al., 2017*). Upon induction with RAP in 1h-old ring stages, CB-EME (*Figure 2*, *Video 7*) and CB-HALO (*Figure 1—figure supplement 1B,C*) labelled F-actin structures completely disappeared in late trophozoites and schizonts and closely resembled parasites treated with cytochalasin-D (*Figure 1C*). As previously reported (*Das et al., 2017*), PfACT1-disrupted parasites could not invade new erythrocytes (*Figure 2B*(ii)). We observed a ≈ 10x reduction in emerald signal upon disruption of PfACT1 (*Figure 2C*), which could be due to the proteasomal degradation of actin-chromobodies, when they are not bound to actin, as was seen by others for a different chromobody (*Tang et al., 2016*).

We previously reported that apicoplast inheritance depends on PfACT1 (*Das et al., 2017*). In order to determine the localisation of F-actin during apicoplast segregation we used deconvolution microscopy on fixed parasites stained with the apicoplast marker CPN60, which revealed apicoplasts to be in close proximity with F-actin structures (*Figure 2D,E*; *Figure 2—figure supplement 1B* lower panel). A Pearson's R value of 0.4 was obtained for colocalisation of the apicoplast(s) signal with the CB-EME channel in *Figure 2—figure supplement 1B* lower panel, as compared to 0.1 with the DAPI channel. We also scored for individual apicoplast signals which are proximal to the F-actin network (example shown in *Figure 2E*) and found 75% ± 7% apicoplasts to lie apposed to the F-actin network (N = 445). Upon disruption of PfACT1, a defect in apicoplast segregation was apparent (*Figure 2D* PfAct1cKO), which recapitulated the phenotype previously observed (*Das et al., 2017*).

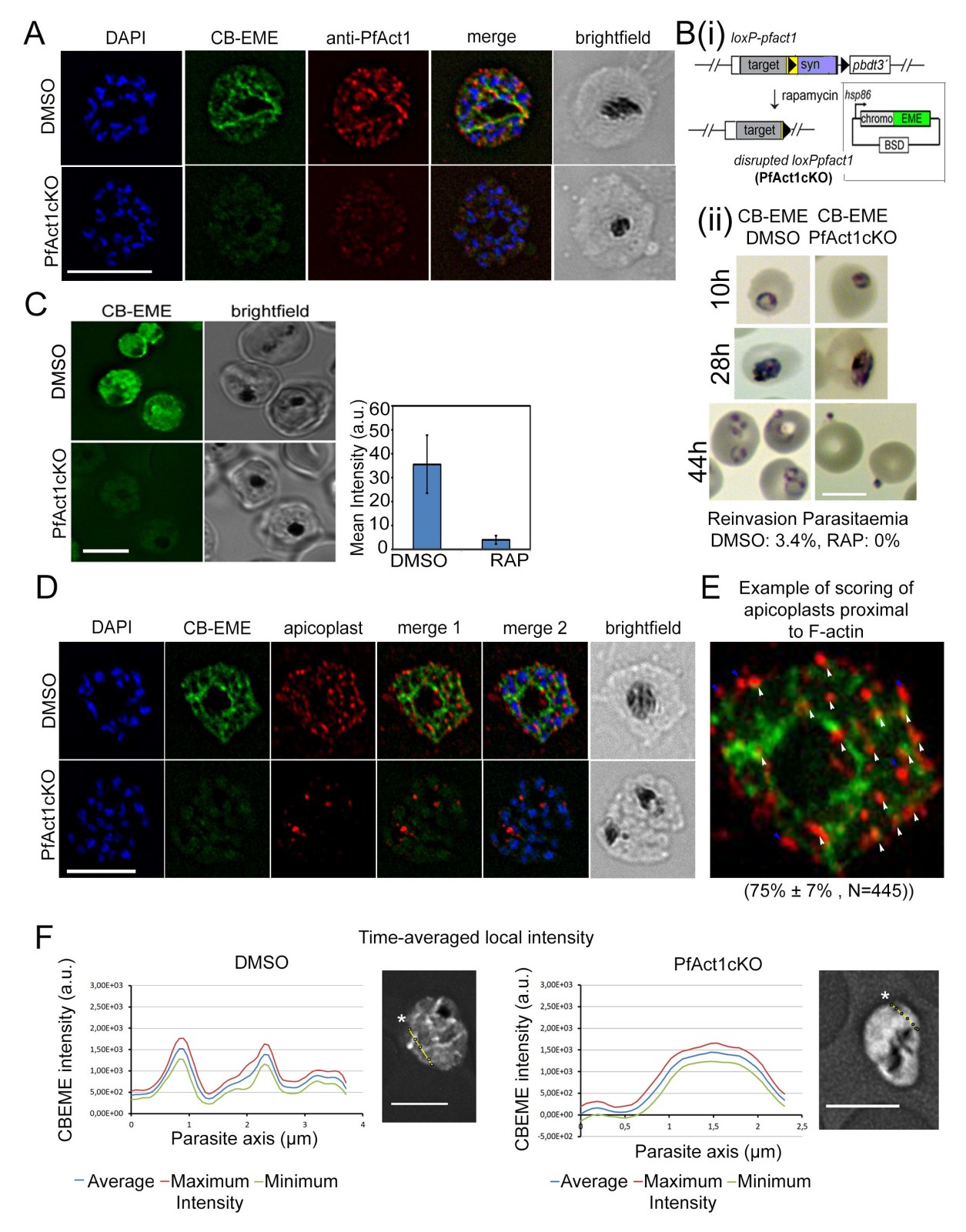

**Figure 2.** The CB-EME-labelled F-actin network is disrupted upon genetic excision of *pfact1*. (**A**) IFA showing the F-actin network in *P. falciparum* (CB-EME), which stains the same network as an actin antibody (anti-PfAct1), though at slightly different regions, Pearson's R value = 0.6, compared to R values of 0.2 when the CB-EME signal was tested for colocalisation with the DAPI signal. DAPI labels nuclei. (**B**) (**i**). Schematic of transfection of pB-CB-EME into the loxPpfact1 strain and PfACT1 loss upon rapamycin (RAP)-mediated DiCre-activation. (**ii**) Giemsa-stained parasites showing time points

*Figure 2 continued on next page*

*Figure 2 continued*

after induction with RAP. Invasion is abrogated in PfACT1 cKOs as compared to the DMSO controls (lower panels). (**C**) Stills from live imaging of CB-EME-expressing parasites and the loss of fluorescence intensity upon RAP-treatment. Right panel shows quantification of fluorescence intensities. See also *Video 7*. (**D**) IFA showing apicoplasts (red) in context of the F-actin network (CB-EME, DMSO) and the disruption of the network together with apicoplasts when PfACT1 is deleted (PfAct1cKO). (**E**) Example of scoring of apicoplast signals proximal to CB-EME signal (white arrows) and not proximal to the CB-EME signal (blue arrows). 445 data points were collected from three repeat experiments. The mean and SD have been depicted below the image. (**F**) Quantification of intracellular F-actin dynamics: Time-averaged local intensity along a defined transect (yellow line) shows defined areas of F-actin accumulation as discrete peaks (left panel) as compared to no peaks along a transect in the PfACT1 cKO. Asterisks indicate the start of the measurement axis (0 µm). Scale bar 5 µm. Additional transects confirm these differences in *Figure 2—figure supplement 2*.

DOI: https://doi.org/10.7554/eLife.49030.013

The following figure supplements are available for figure 2:

**Figure supplement 1.** Super-resolution of the CB-EME-labelled F-actin network.
DOI: https://doi.org/10.7554/eLife.49030.014
**Figure supplement 2.** Quantification of intracellular F-actin dynamics (additional transects).
DOI: https://doi.org/10.7554/eLife.49030.015

In contrast, no obvious defects in mitochondria segregation could be detected in PfACT1-disrupted parasites (*Figure 2—figure supplement 1C* and *Video 7*) as previously reported (*Das et al., 2017*), implying that unlike apicoplasts, mitochondria do not require F-actin for migration into daughter cells.

Next, we quantified spatiotemporal accumulation of F-actin in *P. falciparum* parasites (*Figure 2F*). In order to do so, we performed time-averaged local intensity profiling on acquired time-lapse videos. This enabled us to measure localised fluctuations in F-actin along defined transects in various positions in the cell over time (yellow line, *Figure 2F*). Stable time-averaged F-actin peaks were observed, which could be easily distinguished from background and from the signal in PfACT1KO parasites (*Figure 2F*, *Figure 2—figure supplement 2*). We observed the highest time-averaged intensity peaks around the food vacuole, as also noted by super-resolution microscopy (*Figure 2—figure supplement 2*, lower left panel).

We reasoned that, since most canonical actin filament stabilising and nucleating proteins are absent in Apicomplexa, the parasite must depend on formins for F-actin assembly. Previously, PfFRM1 has been localised to the invasion junction and PfFRM2 to the cytosol (*Baum et al., 2008b*). Since we observed the intracellular F-actin network in the cytosol, we speculated that Formin-2 is the main nucleator of F-actin during intracellular parasite development.

## Apicomplexan Formin-2 sequences contain a PTEN-C2-like domain found usually in plant formins

Formins possess a formin homology (FH) one and an FH2 domain, which nucleate actin monomers as well as elongate unbranched F-actin by continuous processive binding to the barbed end of the filament (*Courtemanche, 2018*). In a previous report (*Baum et al., 2008b*), only FH1/FH2 domains were described for apicomplexan formins. Here, we queried for presence of known PFAM domains using NCBI conserved domain search and in addition to FH1/FH2, found tetra-tricopeptide repeat (TPR) domains in both PfFRM1 and TgFRM1, while a PTEN C2-like domain was recognised in PfFRM2 and TgFRM2 (*Figure 3A*). This led us to hypothesise that Formin-1 and Formin-2 with different N-terminal

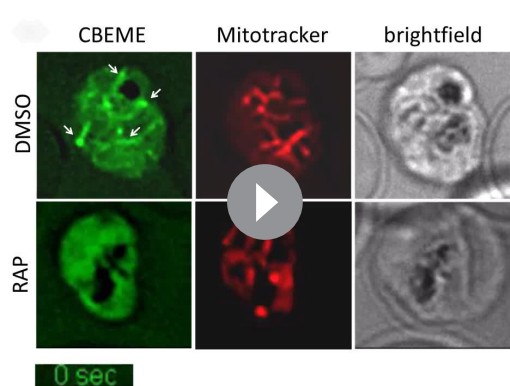

**Video 7.** CB-EME staining disappears upon conditional genetic deletion of *pfact1*. Ring stage LoxPpfACT1/CBEME parasites were pulse treated with DMSO or RAP for 4 hr and imaged after 40 hr. CB-EME was imaged in the green channel and shows a disappearance of F-actin upon RAP-treatment. Mitochondria were stained with Mitotracker (red channel). Acquisition time is shown in seconds. Scale bar 5 µm.
DOI: https://doi.org/10.7554/eLife.49030.016

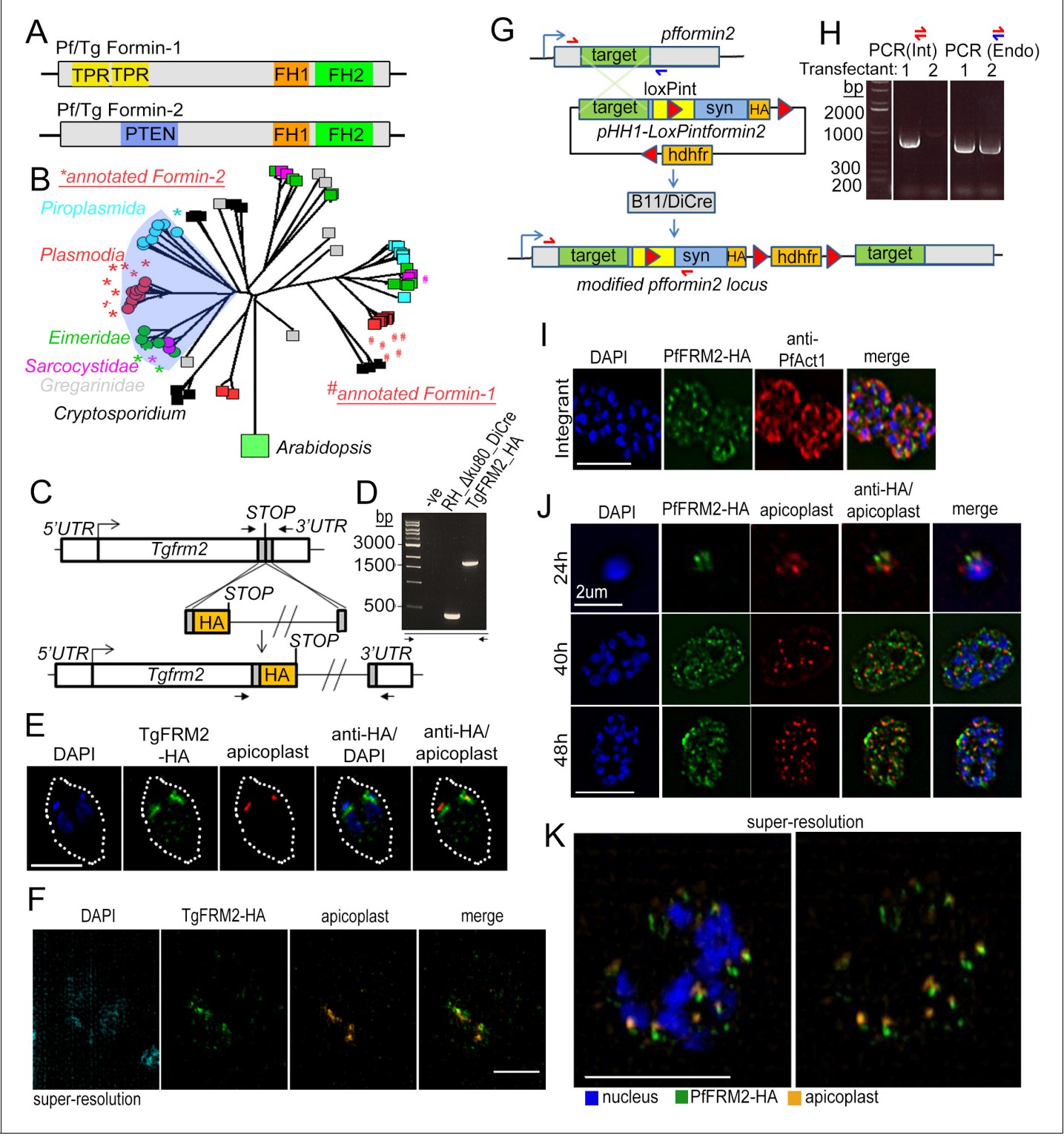

**Figure 3.** Apicomplexan formins have distinct protein domains, with Formin-2 localising to apicoplasts in *Toxoplasma* and *P. falciparum*. (A) Other than the conserved FH1/FH2 domains common to all formins, Pf and Tg Formin-1 contain tetratricopeptide repeat domains (TPR), while Pf and Tg Formin-2 contain a PTEN C2-like domain. (B) Rooted neighbour-joining tree of FH2 domains detected in apicomplexan sequences flagged by hmmsearch and extracted from alignments produced by hmmalign, both using the PFAM profile PF02181.23: Proteins with sub-sequences similar to PTEN-C2 domains (detected by psi-Blast) are indicated with circular leaf symbols (within the shaded blue area). Those sequences already annotated as Formin-1 (#) and Formin-2 (*) are indicated. Colour coding of the leaf nodes: Red: Plasmodium, Green: Eimeria, Magenta: Sarcocystidae, Cyan: Piroplasmida, Black:

*Figure 3 continued on next page*

*Figure 3 continued*

Cryptosporidium, Grey: Gregarinidae (C) Strategy depicting endogenous C-terminal HA tagging of *tgfrm2* in Toxoplasma. CRISPR/Cas9 was exploited to introduce a double-stranded DNA break and repair DNA amplified by PCR with homologous DNA regions coding for 3xHA. (D) Diagnostic PCR confirming integration of DNA described in C into the RH_Δku80_DiCre line. (E) IFA showing localisation of TgFRM2-HA (green) at the vicinity of the apicoplast staining (anti-G2Trx, red). Nuclei are stained with DAPI (blue). White dotted line depicts the parasite vacuole outline. Parasites were grown for 24 hr. Scale bar 5 µm F. Super-resolution microscopy confirming the close apposition of TgFRM2-HA (green) to the apicoplast (anti-G2Trx, orange). Toxoplasma parasites were fixed 24 hr after inoculation. Scale bar 2.5 µm. (G) Strategy showing simultaneous floxing and C-terminal HA tagging of the endogenous *pffrm2* locus using single cross over recombination into a DiCre expressing strain to give rise to the LoxPpfformin2 strain (modified). A 906 bp targeting sequence (target) followed by a heterologous intron with an internal *LoxP* site (LoxPint) followed by synthetic recodonised DNA sequence with additional LoxP sites at the 3′ end ensures recombination upstream of LoxP sites. Human dihydrofolate reductase (hdhfr) confers resistance to the drug WR99210. Primers for diagnostic PCR have been annotated as half arrows. (H) Diagnostic PCR on genomic DNA from two transfectants confirmed integration in one of the two transfected lines (Transfectant 1). Primers depicted with red half arrows (int) are specific to the integrated locus, while red and blue half arrows (Endo) are specific to the endogenous locus (I) IFA showing localisation of PfFRM2-HA (green) in the context of a PfACT1-antibody staining (red). Nuclei are stained with DAPI (blue). (J) IFA showing localisation of PfFRM2-HA in context of apicoplasts using a CPN60 antibody (red) throughout *P. falciparum* intracellular development (24, 40, 48 hr). (K) Super-resolution image confirming the apparent proximity of PfFRM2-staining (green) with apicoplasts (orange). Scale bars are 5 µm, except where otherwise noted.

DOI: https://doi.org/10.7554/eLife.49030.017

The following figure supplement is available for figure 3:

**Figure supplement 1.** Positioning of TgFRM2 with respect to apicoplasts.

DOI: https://doi.org/10.7554/eLife.49030.018

domains diverged early in evolution and different domain organisations have been retained for different functions. We queried for FH2-domain containing proteins from various apicomplexans and found that Formin-2-like sequences are found in a different clade from Formin-1-like sequences (*Figure 3B*), as also previously noted (*Baum et al., 2008b*). Strikingly, the PTEN-C2-domain (or a diverged PTEN-C2 domain) was found only in Formin-2-like sequences (*Figure 3B*). Interestingly, PTEN-C2 domains are important for membrane recruitment (*Das et al., 2003*) and a class II rice Formin uses this domain to be recruited to chloroplast membranes (*Zhang et al., 2011*), leading us to hypothesise that a similar mechanism operates for apicoplast recruitment of Formin-2 sequences in apicomplexans.

### *Plasmodium* and *Toxoplasma* Formin-2 localise adjacent to apicoplasts

In order to characterise the role of Formin-2 within the evolutionary niche of apicomplexans, we decided to perform a comparative analysis in both *T. gondii* and *P. falciparum*. Therefore, we epitope tagged Formin-2 in both parasites. For tagging in *T. gondii* we used a CRISPR/Cas9-based strategy to introduce a 3x hemagglutinin (3 HA) tag at the TgFRM2 C-terminus (*Figure 3C*) and confirmed correct integration by diagnostic PCR (*Figure 3D*). Upon co-staining with the anti-apicoplast antibody G2-Trx (Biddau and Sheiner, unpublished), we found TgFRM2 to be localised adjacent to apicoplast(s) (*Figure 3E*), which was confirmed by super-resolution microscopy (*Figure 3F*). Upon quantification of co-localisation (*Figure 3—figure supplement 1*), we found 30% apicoplasts partially colocalised and 58% apicoplasts adjacent to and in contact with the TgFRM2 signal (n = 142). For localisation of PfFRM2, we simultaneously epitope tagged and floxed PfFRM2 by single cross-over homologous recombination in a DiCre-expressing parasite strain (*Figure 3G*) and confirmed integrants by diagnostic PCR (*Figure 3H*). Integrants were cloned by limiting dilution and two distinct clones of 'LoxPpfformin2' were used for phenotypic characterisation. PfFRM2 showed a punctate pattern within cells (*Figure 3I*). Next, we checked for PfFRM2 localisation in relation to apicoplasts and observed a close proximity of the apicoplasts with most of the PfFRM2 punctae throughout the 48 hr *Plasmodium* blood stage life-cycle (*Figure 3J*), which was confirmed by super-resolution microscopy (*Figure 3K*). Upon quantification of co-localisation of the apicoplast signal with that of the formin punctae, we obtained a Pearson's R value of 0.55 ± 0.2 (N = 4), the high variance implying that the close proximity of the apicoplasts to Formin-2 may be dynamic. In conclusion, both *Toxoplasma* and *P. falciparum* Formin-2 (dynamically) localise in close proximity to apicoplasts.

## DiCre-mediated conditional disruption of Formin-2 causes a defect in apicoplast segregation in *P. falciparum*

Next, we wished to evaluate the fate of *P. falciparum* upon conditional DiCre-mediated disruption of the *pffrm2* gene (*Figure 4A*). 1 hr old, tightly synchronised ring stage parasites were divided into two flasks and either pulse-treated with RAP or DMSO (control) for 4 hr and their phenotype determined 44 hr post RAP-treatment. Excision was determined by diagnostic PCR of the genomic locus (*Figure 4B*) and fitness of the PfFRM2 conditional knock out (cKO) was measured by a growth curve which showed significant loss of viability (*Figure 4C*). Loss of protein was ~90% (averaged from three independent experiments) as determined by Western blot (*Figure 4D*) and was confirmed by IFA (*Figure 4E*), which indicated a loss of protein in ~95% parasites (N = 350). Giemsa stained PfFRM2 cKO parasites were dysmorphic with apparent inclusions of haemoglobin (*Figure 4F*, red arrows). In order to determine the morphological defects in PfFRM2 KO parasites, we co-stained PfFRM2 cKO parasites with several organellar markers and were unable to see significant differences (not shown), except for apicoplast segregation (*Figure 4G*). The number of parasites with normally segregated apicoplasts was significantly reduced, with a high percentage of cells showing collapsed or morphologically aberrant apicoplasts (*Figure 4G,H*). A range of apicoplast phenotypes was evident, from totally collapsed, intermediate to apparently normal (*Figure 4G,H*). To determine if the loss of viability of the PfFRM2 cKO parasites was solely due to loss of the apicoplast, we attempted to rescue the phenotype with 200 µM isopentenyl pyrophosphate (IPP) which has been previously shown to complement growth in parasites lacking apicoplasts (*Yeh and DeRisi, 2011*). However, we did not see any improvement in viability (*Figure 4I*), indicating that the loss of fitness is due to additional defects caused by abrogation of F-actin dynamics in the parasite, which we shall address shortly in Section 5.

## DiCre-mediated disruption of Formin-2 abrogates the actin network in *P. falciparum* schizonts

Apicoplast inheritance is critically dependent on PfACT1 (*Das et al., 2017*). Hence, we subsequently determined whether F-actin assembly and dynamics are interfered upon deletion of PfFRM2. We expressed CB-EME in the LoxPpfformin2 strain to generate the line LoxPpfformin2/CBEME (*Figure 5A*) and visualised actin filaments by IFA and by live time lapse microscopy. IFA analysis showed PfFRM2 punctae localised primarily with CB-EME-labelled filamentous structures (Pearson's R value = 0.36, as compared to 0.02 with the DAPI channel) (*Figure 5B*, DMSO and *Figure 5—figure supplement 1*). When we manually counted the individual PfFRM2-HA punctae, we found 80% ± 6% closely apposed to the F-actin signal (N = 200). Upon DiCre-mediated excision in ring stages, we saw a complete abrogation of the dynamic F-actin network in mature schizont stage parasites (*Figure 5B* RAP), which dropped from exhibiting an F-actin network in 92 ± 5% cells in WT to 5 ± 4% cells in PfFRM2 cKO (*Figure 5C,D*). Surface intensity plots showed a dramatic reduction in F-actin peaks in the PfFRM2 cKO parasites (*Figure 5E,F*). Furthermore, we confirmed the apicoplast inheritance phenotype in PfFRM2 cKO parasites expressing CB-EME (*Figure 5G*). Since IPP could not rescue the growth defect in PfFRM2 cKO parasites (*Figure 4I*) and a dramatic disruption of F-actin signal was observed, additional defects due to loss of F-actin nucleation were subsequently investigated. We noticed a faint emerald signal inside nuclei of fixed PfFRM2 cKO parasites (*Figure 5B*). Since this signal is absent in live microscopy, we speculate that this is due to a potential bleed-through from the blue to the green channel in fixed parasites.

## DiCre-mediated conditional disruption of Formin-2 affects daughter cell formation/cytokinesis in *P. falciparum*

As a first step towards characterisation of additional defects, we determined the number of nuclei in PfFRM2 cKO parasites 40 hr post RAP-treatment, and found a significant decrease in the number of nuclei in these parasites, as compared to the DMSO control (*Figure 6A*), indicating a defect in development or schizogony. Since PfACT1 is required for normal cytokinesis (*Das et al., 2017*), we examined if the IMC is normally formed in PfFRM2 cKO parasites. In an attempt to exclude younger, trophozoite-stage parasites, we purified mature schizonts on a 70% Percoll cushion and determined by IFA using GAP45 as a marker if IMC formation was compromised in these parasites. Fully segmented IMC staining dropped from 58 ± 8% in WT to 19 ± 8% in PfFRM2 cKOs (*Figure 6B,C*). When

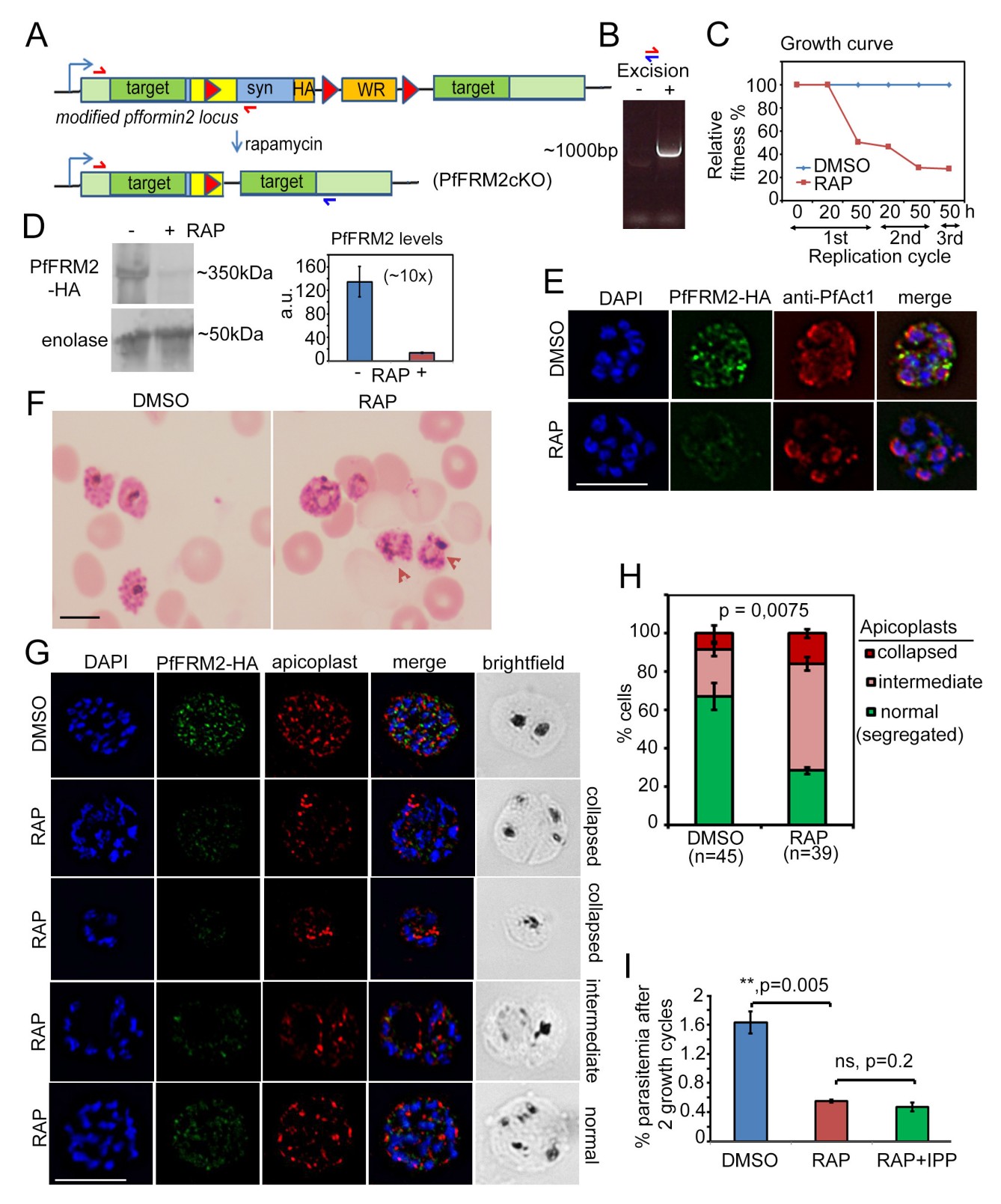

**Figure 4.** Conditional excision of *pffrm2* disrupts apicoplast segregation and causes a severe fitness defect in *P. falciparum*. (**A**) Strategy showing the DiCre-mediated genomic excision of the LoxPpfFRM2 locus (modified). Primers for diagnostic PCR have been annotated as red/blue half arrows. (**B**) Diagnostic PCR confirming genomic excision of the *pffrm2* locus upon rapamycin treatment (+). (**C**) A growth curve showing the relative fitness of RAP-treated PfFRM2 cKO parasites in comparison to DMSO controls. Various time points (h) from the pulse treatment of 1h-old rings at time 0 in the 1st, 2nd

*Figure 4 continued on next page*

*Figure 4 continued*

and 3$^{rd}$ replication cycles have been measured. (**D**) left panel, Western blot showing the loss of PfFRM2-HA upon RAP-treatment, enolase has been used as a control. Right panel, Quantification of PfFRM2-HA protein levels using intensity values normalised to enolase from three different immunoblots shows at least a 10-fold drop in protein levels, Error bars depict SD. Values are in arbitrary units (a.u.) (**E**) IFA showing loss of PfFRM2-HA staining (green) upon RAP-treatment. Levels of PfACT1-staining (red) do not change. (**F**) Giemsa-stained images of RAP-treated parasites reveal dysmorphic parasites 44 hr after RAP-treatment. (**G**) Apicoplast staining (red) is affected to various degrees – collapsed, intermediate and apparently normal in RAP-treated parasites as compared to DMSO controls, where a normal punctate staining for apicoplasts is visible in a multi-nucleated schizont. (**H**) Quantification of phenotypes seen in G) shows a 3-fold reduction in normal apicoplast staining in PfFRM2 cKOs (RAP). Error bars depict SD. (**I**) Isopentenyl pyrophosphate (IPP) cannot rescue the fitness defect (RAP +IPP) in PfFRM2 KO parasites (RAP) compared to DMSO controls, as measured by final parasitemia after two growth cycles. Error bars depict SD. Scale bars 5 μm.
DOI: https://doi.org/10.7554/eLife.49030.019

we allowed these Percoll-purified mature PfFRM2 cKO parasites to egress and compared them to control parasites, we found conjoined merozoites in PfFRM2 cKOs (*Figure 6D,E*), a defect previously seen in PfACT1 cKO parasites (*Das et al., 2017*), indicating that PfFRM2 and PfACT1 coordinate cytokinesis in *P. falciparum*. We have, however, not followed the fate of the excluded 'younger' parasites with fewer nuclei in the PfFRM2 cKO population. We cannot therefore rule out a function for Formin-2 earlier in schizogony, e. g. in endocytosis, in addition to a role in cytokinesis.

Since PfFRM1 was localised to the parasite apex/invasion junction during host cell entry (*Baum et al., 2008b*), we reasoned that apical polymerisation of F-actin should not be affected in PfFRM2 cKO parasites, if indeed the two formins perform distinct functions in their distinct localisations. To this end, we allowed mature schizonts to egress and release free merozoites and immediately imaged them by live fluorescence microscopy. Consistent with this hypothesis, we found that the ability of F-actin polymerisation at the parasite apex was not compromised in PfFRM2 cKO parasites (*Figure 6F*), strongly indicating distinct functions of PfFRM1 and PfFRM2.

Despite showing a significant growth defect, PfFRM2 cKO parasites did reinvade RBCs and established ring stage parasites. In good agreement with the observations above and a general role of PfFRM2 in F-actin nucleation during intracellular parasite development, we could not observe any F-actin dynamics in early trophozoite stage parasites or peripheral accumulations of F-actin in these parasites (*Figure 6—figure supplement 1*).

## Conditional deletion of Formin-2 in *Toxoplasma* disrupts apicoplast segregation and F-actin dynamics

Finally, in order to assess if the function of Formin-2 is conserved in apicomplexan parasites, we analysed its role in *T. gondii*. We first checked the localisation of TgFRM2 with respect to the F-actin network. Similar to *P. falciparum*, TgFRM2-HA formed punctae on the CB-EME labelled F-actin network within the parasite. There it appeared to co-localise with a polymerisation centre (*Figure 5—figure supplement 1Tg*), recently also described in an independent report (*Tosetti et al., 2019*). Next, we simultaneously floxed *tgfrm2* together with addition of a C-terminal YFP tag to create the LoxPTgFRM2 line (*Figure 7A*). This enabled us to confirm localisation of Formin-2 and determine the comparative effect of a conditional knock out of Formin-2 in *Toxoplasma*. Integration of the C-terminal YFP-tag and *LoxP* sites was confirmed by diagnostic PCR, as was excision of the *tgfrm2* locus upon RAP-treatment (*Figure 7—figure supplement 1A*). For the localisation of TgFRM2 it was necessary to stain fixed parasites with a YFP-antibody, suggesting low expression levels of TgFRM2 (*Figure 7—figure supplement 1B*). We confirmed

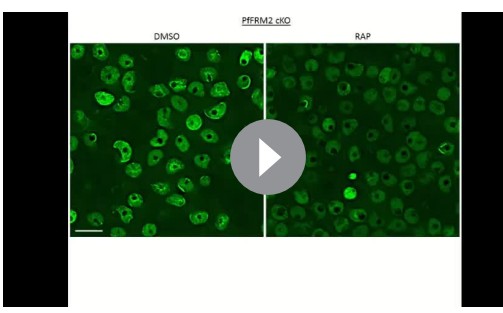

**Video 8.** Actin filaments disappear upon genetic deletion of *pffrm2*. Ring stage LoxPpfFRM2/CBEME parasites were DMSO- or RAP-treated for 4 hr and imaged 40 hr later. CB-EME was imaged in the green channel and shows a disappearance of intracellular F-actin upon RAP-treatment. Acquisition time is shown in seconds. Scale bar 5 μm.
DOI: https://doi.org/10.7554/eLife.49030.011

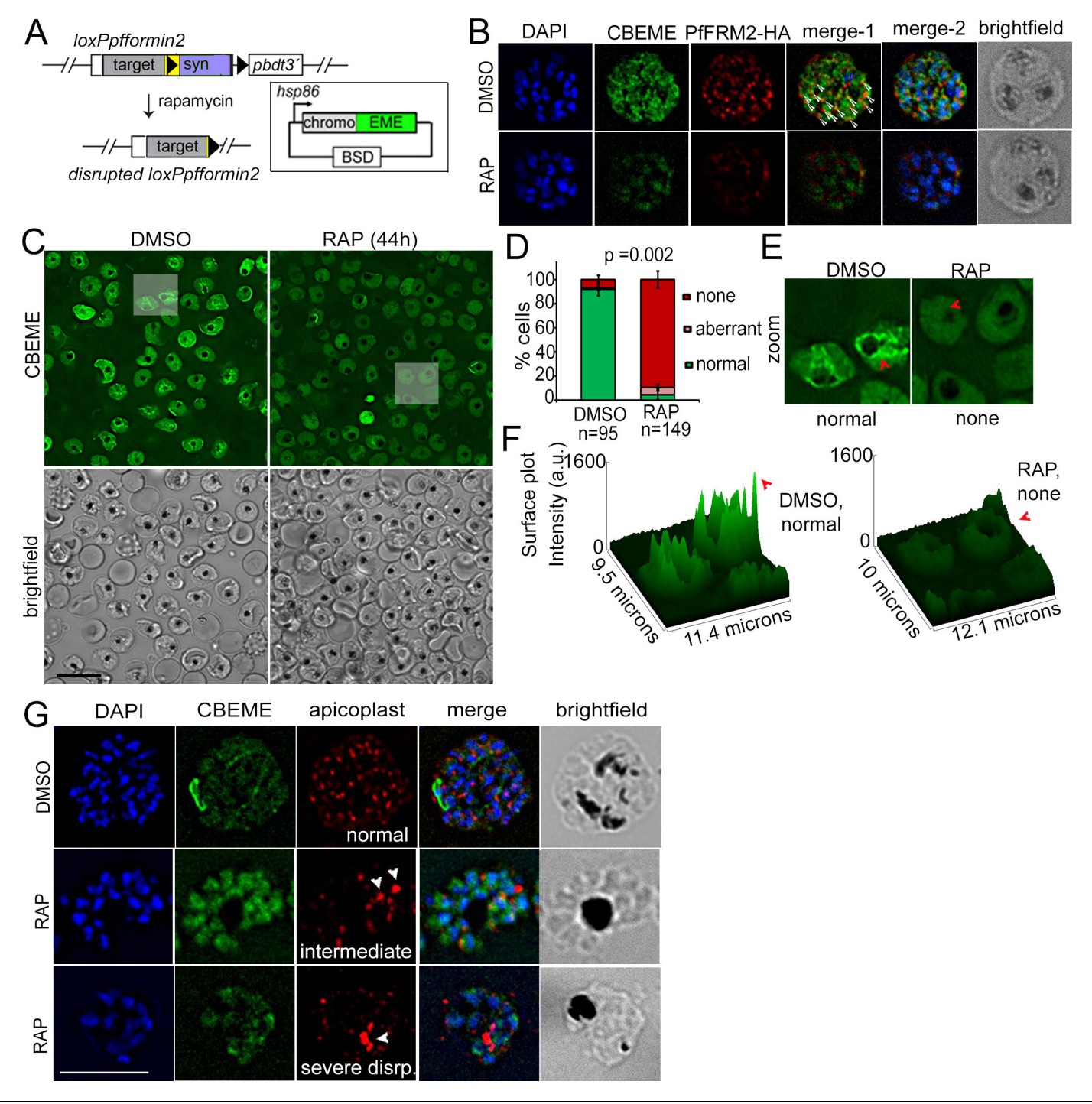

**Figure 5.** Conditional deletion of PfFRM2 abrogates the intracellular F-actin network in *P. falciparum*. (**A**) Strategy showing expression of p-CB-EME in the RAP-inducible LoxPpffrm2 background. (**B**) IFA showing PfFRM2-HA staining in the context of the F-actin network labelled by CB-EME expression (DMSO) and the subsequent loss of CB-EME and PfFRM2-HA staining in PfFRM2 cKO parasites (RAP). (**C**) Stills from a time-lapse movie showing loss of normal intracellular F-actin fluorescence (green). Brightfield images have been provided below. See also *Video 8*. (**D**) Graph showing loss of normal F-actin fluorescence in ~95% RAP-treated parasites. > 90% of DMSO controls show presence of the network. (**E**) Zoomed images of indicated boxed parasites in (**C**) showing loss of the actin network in RAP (none, red arrows) as compared to DMSO controls (normal). (**F**) Intensity surface plots clearly show a difference in localised intensity (red arrows) within cells on comparing the DMSO-control parasites with RAP-treated PfFRM2 cKOs. (**G**) IFA staining of the apicoplast with a CPN60 antibody (red) together with the fluorescent F-actin network (green) confirms a defect in apicoplast segregation

*Figure 5 continued on next page*

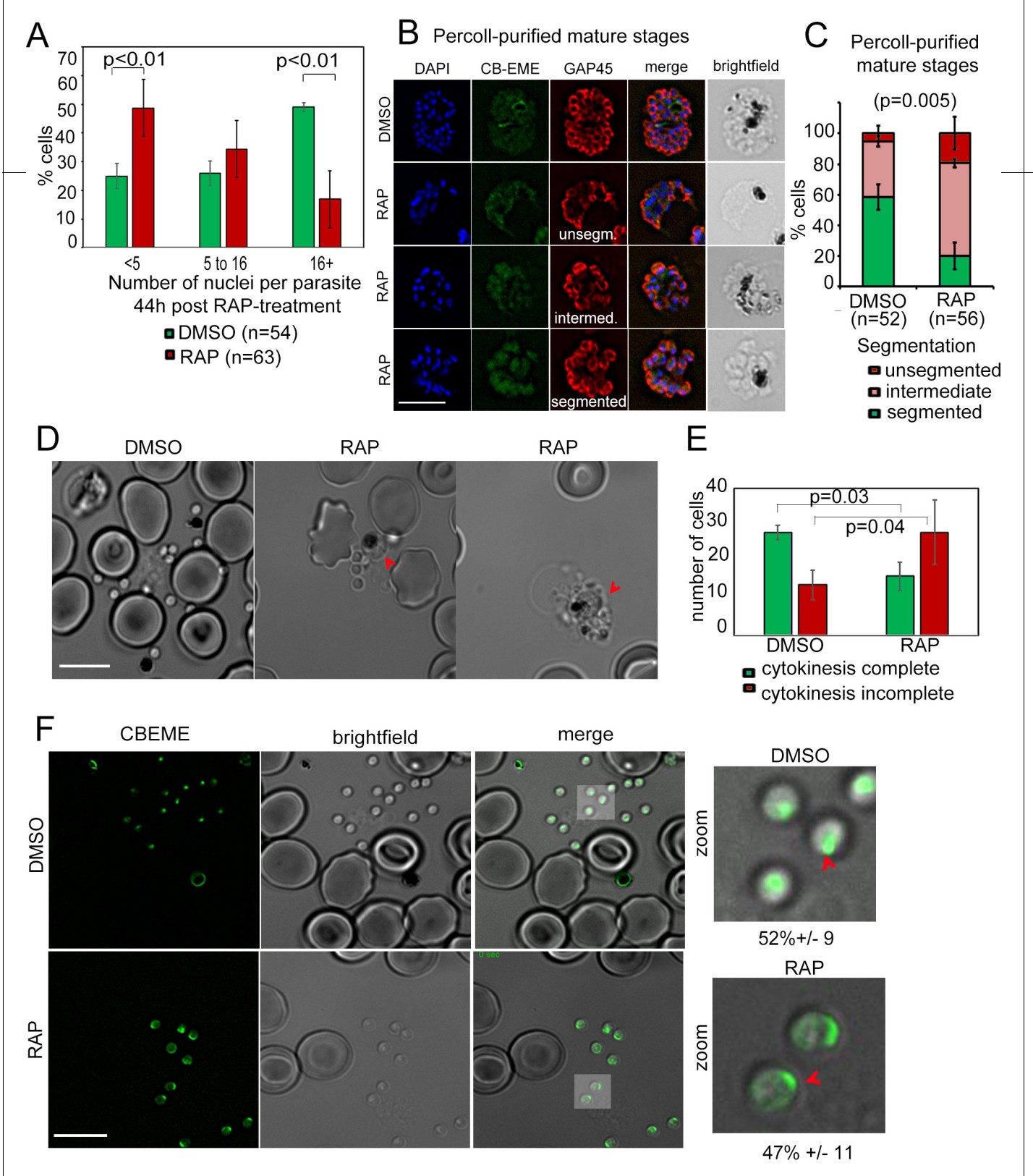

**Figure 6.** Defects in merozoite formation/cytokinesis in PfFRM2 cKO parasites. (**A**) Quantification by IFA of number of nuclei in DMSO or RAP-treated LoxPpffrm2/CBEME parasites, 40 hr post treatment: Number of nuclei were binned to <5, 5 to 16 and 16 + in DMSO controls and RAP-treated parasites. The graph shows a significant reduction in the number of DAPI-stained nuclei in RAP-treated parasites. (**B**) IFA on Percoll-purified mature

*Figure 6 continued on next page*

*Figure 6 continued*

LoxPpffrm2/CBEME schizonts 44 hr post DMSO/RAP treatment, further allowed to mature for 4 hr in Compound 2: An anti-GAP45 antibody (red) revealed defects to varying degrees in IMC formation in these parasites. Examples for unsegmented, intermediate and segmented IMCs have been provided. (C) Quantification of defects in IMC formation from the IFA in B shows a significant reduction in segmented daughter merozoite formation in the RAP-treated population. (D) When DMSO/RAP-treated schizonts were allowed to egress, conjoined merozoites around the FV were apparent in the RAP-treated populations (red arrows) much more frequently than the DMSO controls. (E) Quantification of phenotypes observed in D revealed >2 times as many PfFRM2 cKOs parasites not completing cytokinesis as compared to DMSO controls. (F) Post-schizont egress, merozoites from DMSO controls and RAP-treated group show similar propensity to polymerise apical F-actin (CB-EME fluorescence shown in green). Red arrows show apical F-actin in zoomed images (right panels). Scale bars 5 μm.
DOI: https://doi.org/10.7554/eLife.49030.022

The following figure supplement is available for figure 6:

**Figure supplement 1.** Loss of peripheral F-actin in LoxPpfFRM2/CB-EME trophozoites in the second growth cycle (72 hr post RAP-treatment).
DOI: https://doi.org/10.7554/eLife.49030.023

localisation of TgFRM2 adjacent to the apicoplast (*Figure 7B* upper panel). Upon RAP-treatment, excision of TgFRM2 was apparent in 36 ± 4% (n = 300) of vacuoles, as assessed by quantification of parasites where no TgFRM2 could be detected by IFA. Importantly, loss of TgFRM2 staining correlated with an apicoplast segregation phenotype in 65 ± 7% (n = 300) of parasites (*Figure 7D*). A baseline apicoplast segregation phenotype was observed in 1% (±0; n = 300) of vacuoles in the control population. Loss of TgFRM2 had no impact on mitochondrial replication or morphology (*Figure 7E*). Transient expression of CB-EME in LoxPTgFRM2 parasites enabled us to image F-actin and demonstrated that, in good agreement with data from *P. falciparum*, intracellular F-actin formed a polymerisation centre adjacent to the apicoplast (*Figure 7C* control). Intriguingly, excision of TgFRM2 (*Figure 7C* RAP) led to the disappearance of intracellular F-actin at the polymerisation centre, while (in contrast to *P. falciparum*), the intra-vacuolar F-actin network was still present (*Figure 7C*). In *Toxoplasma*, actin polymerisation within the residual body (and consequently the formation of the intra-vacuolar F-actin network) has been attributed to Formin-3 (*Tosetti et al., 2019*), which is absent in *P. falciparum*.

## TgFRM2 represents a key factor in maintaining spatiotemporal actin dynamics and F-actin flow within intracellular parasites

We further investigated the contribution of Formin-2 to the overall maintenance and dynamics of the F-actin network. For this purpose, we generated a conditional dimerisable-CRISPR/Cas9 system in *T. gondii* that allowed us to rapidly and robustly disrupt genes of interest in CB-EME-expressing

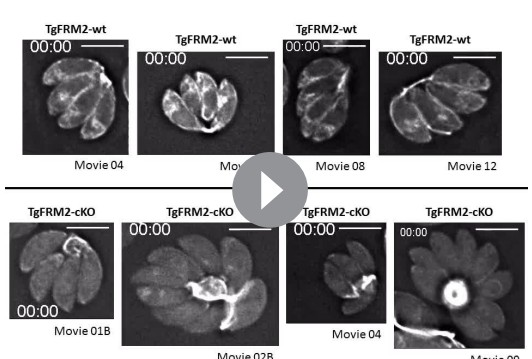

**Video 9.** F-actin dynamics in TgFRM2-wt and TgFRM2-cKO parasites. Time-lapse movie showing CB-EME signal. The movies were captured at a speed of 0.25 s/frame. Scale bars are 5 μm. Movies are depicted at 60fps. Time is shown as mm:ss.
DOI: https://doi.org/10.7554/eLife.49030.029

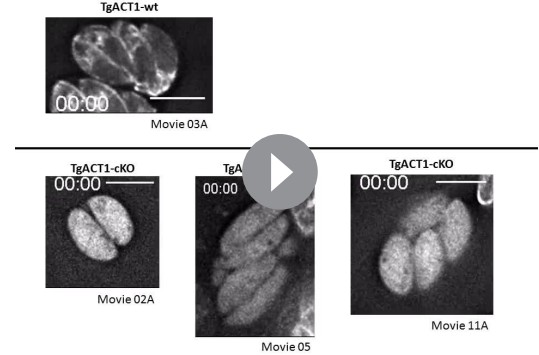

**Video 10.** F-actin dynamics in TgAct-wt and TgAct-cKO parasites. Time-lapse movie showing CB-EME signal. The movies were captured at a speed of 0.29 s/frame. Scale bars are 5 μm. Movies are depicted at 60fps. Time is shown as mm:ss.
DOI: https://doi.org/10.7554/eLife.49030.030

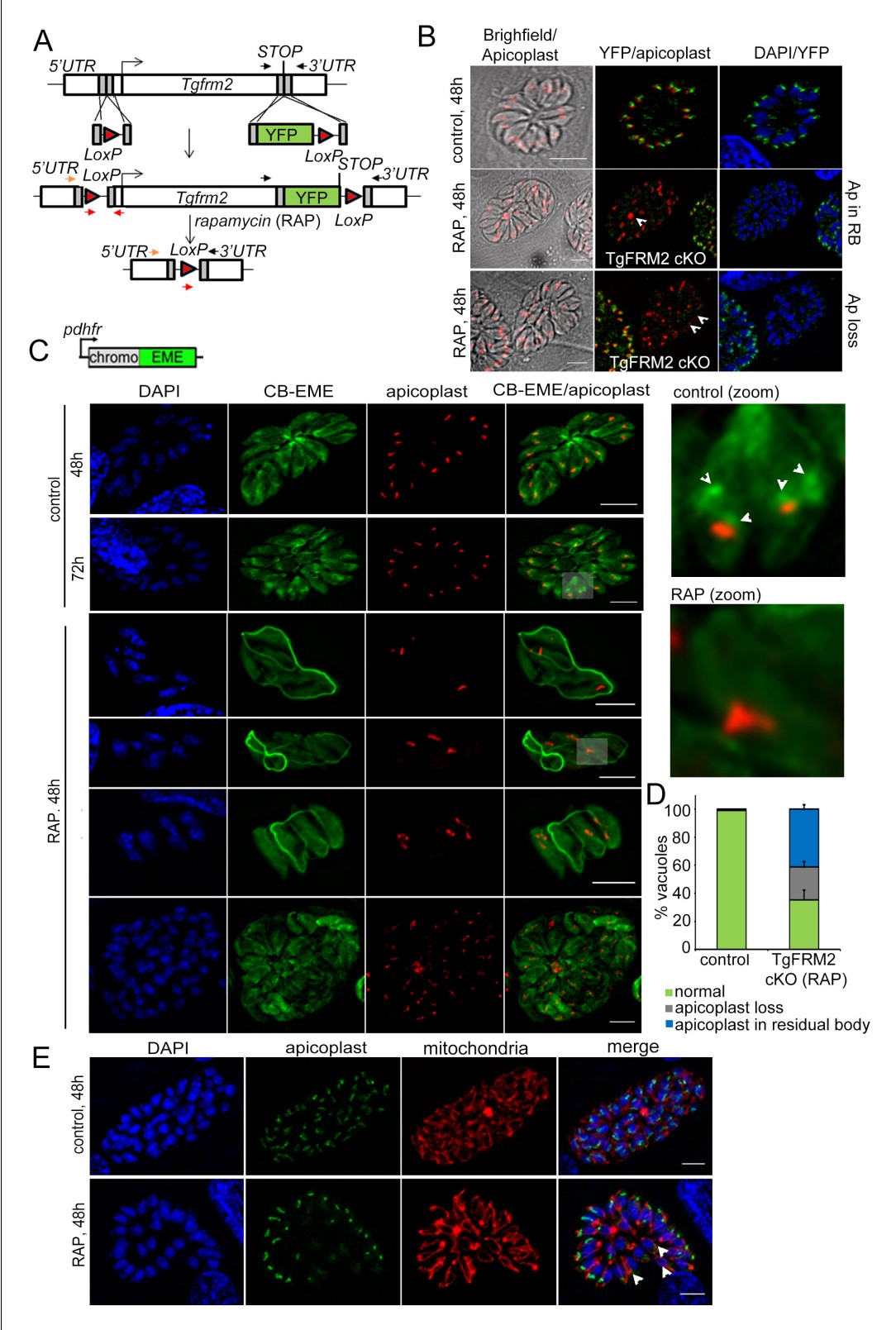

**Figure 7.** Conditional deletion of TgFRM2 disrupts normal segregation of apicoplasts together with abrogation of the intracellular F-actin polymerisation centre. (**A**) Strategy to generate LoxPTgFRM2-YFP, a floxed and C-terminal YFP-tagged *Tgfrm2* locus in the RH_Δku80_DiCre line. For this purpose, CRISPR/Cas9 was exploited to introduce DNA double-strand breaks in the 5' UTR and C-terminus of the *tgfrm2* gene. Integration was confirmed by PCR (see *Figure 7—figure supplement 1*). Arrows represent PCR primers used in *Figure 7—figure supplement 1A*. (**B**) IFA staining with

*Figure 7 continued on next page*

*Figure 7 continued*

anti-YFP (TgFRM2-YFP) and anti-Atrx1 (apicoplast) shows an apicoplast segregation defect in TgFRM2-YFP cKO parasites. In control parasites, TgFRM2-YFP localises to the vicinity of the apicoplast (upper panel). The loss of TgFRM2-YFP causes an apicoplast segregation defect (middle and bottom panels, white arrows). The middle and lower panels depict a TgFRM2-YFP cKO vacuole together with a TgFRM2-YFP positive vacuole for comparison. Apicoplast (Ap) phenotypes in TgFRM2-YFP cKO parasites were classified as Ap loss and Ap in residual body (RB). Scale bars 5 μm. (**C**) Upper panel depicts the CB-EME construct used under the dhfr promoter for expression in the LoxPTgFRM2-YFP strain. Lower panel: IFA showing CB-EME and apicoplast (anti-CPN60) in control and RAP-treated LoxPTgFRM2-YFP parasites. In untreated parasites, the apicoplast localises to intracellular actin polymerisation centres (control, white arrows in zoom). Parasites exhibiting TgFRM2 cKO-specific apicoplast phenotype lack intracellular actin polymerisation centres. Zoomed images depict indicated areas. See also *Figure 7—figure supplement 1B*. (**D**) Quantification of apicoplast inheritance defect shows a significant reduction in apicoplast numbers in TgFRM2 cKOs. Classification of the apicoplast phenotype (apicoplast loss or apicoplast in residual body) refer to IFA depicted in **B**. Vacuoles from three independent experiments were examined. For each biological repeat and condition (control or RAP), 100 vacuoles were counted (total n=300 for each condition). Error bars depict SD. (**E**) IFA showing normal mitochondrial staining (red) in TgFRM2 cKO parasites (RAP, 48 hr) which have lost their apicoplasts (white arrows). Control parasites shows normal apicoplast and mitochondria staining (upper panel). DNA was stained with DAPI, apicoplast staining was performed with anti-Atrx1 antibody and mitochondrial staining with anti-TOM40 antibody. Scale bars 5 μm.

DOI: https://doi.org/10.7554/eLife.49030.024

The following figure supplement is available for figure 7:

**Figure supplement 1.** Loss of TgFRM2-YFP upon RAP-treatment.

DOI: https://doi.org/10.7554/eLife.49030.025

parasites simply by addition of RAP (Stortz, Grech et al. in preparation). Videos captured from time-lapse microscopy were used to perform time-averaged intensity profiling in order to measure CB-EME distribution within parasites over time. In wild-type parasites, the highest CB-EME intensities were observed at the apical and posterior poles and anterior to the nucleus (*Figure 8A*, *Video 9*). Additionally, skeletonisation analysis on videos of wild-type parasites showed actin accumulation in the periphery of intracellular parasites (*Figure 8B*, *Video 9*). Live imaging further revealed that the actin polymerisation centre anterior to the nucleus is highly dynamic and frequently interacts with peripheral actin (*Figure 8C*, *Video 9*).

In agreement with observations made using the DiCre-system (*Figure 7C*), disrupting *tgfrm2* using the conditional CRISPR/CAS9 system also resulted in loss of F-actin at the nucleation centre anterior to the nucleus (*Figure 8A* right panel, *Video 9*). Interestingly, the abundance of peripheral actin was also strongly reduced in TgFRM2-cKO parasites (*Figure 8B*, *Video 9*). Consequently, it is conceivable that the TgFRM2-mediated actin polymerisation centre anterior to the nucleus contributes to the parasite peripheral actin pool, making Formin-2 a key player in regulating actin distribution within intracellular parasites. Intriguingly, but

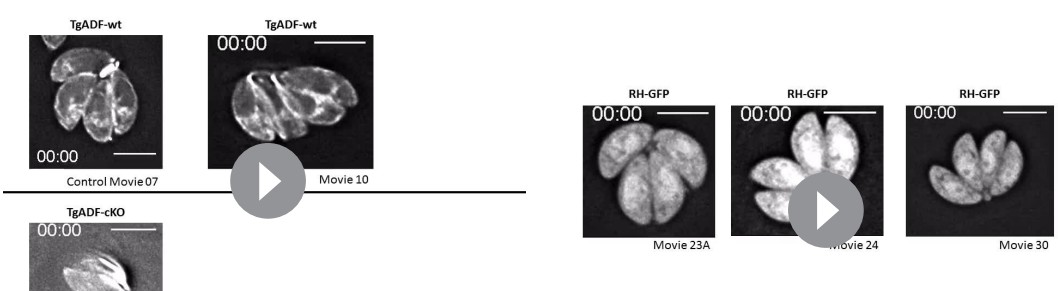

**Video 11.** F-actin dynamics in TgADF-wt and TgADF-cKO parasites. Time-lapse movie showing CB-EME signal. The movies were captured at a speed of 0.32 s/frame. Scale bars are 5 μm. Movies are depicted at 60fps. Time is shown as mm:ss.

DOI: https://doi.org/10.7554/eLife.49030.031

**Video 12.** Live microscopy of RH-GFP parasites. The movies were captured at a speed of 0.33 s/frame. Scale bars are 5 μm. Movies are depicted at 60fps. Time is shown as mm:ss.

DOI: https://doi.org/10.7554/eLife.49030.032

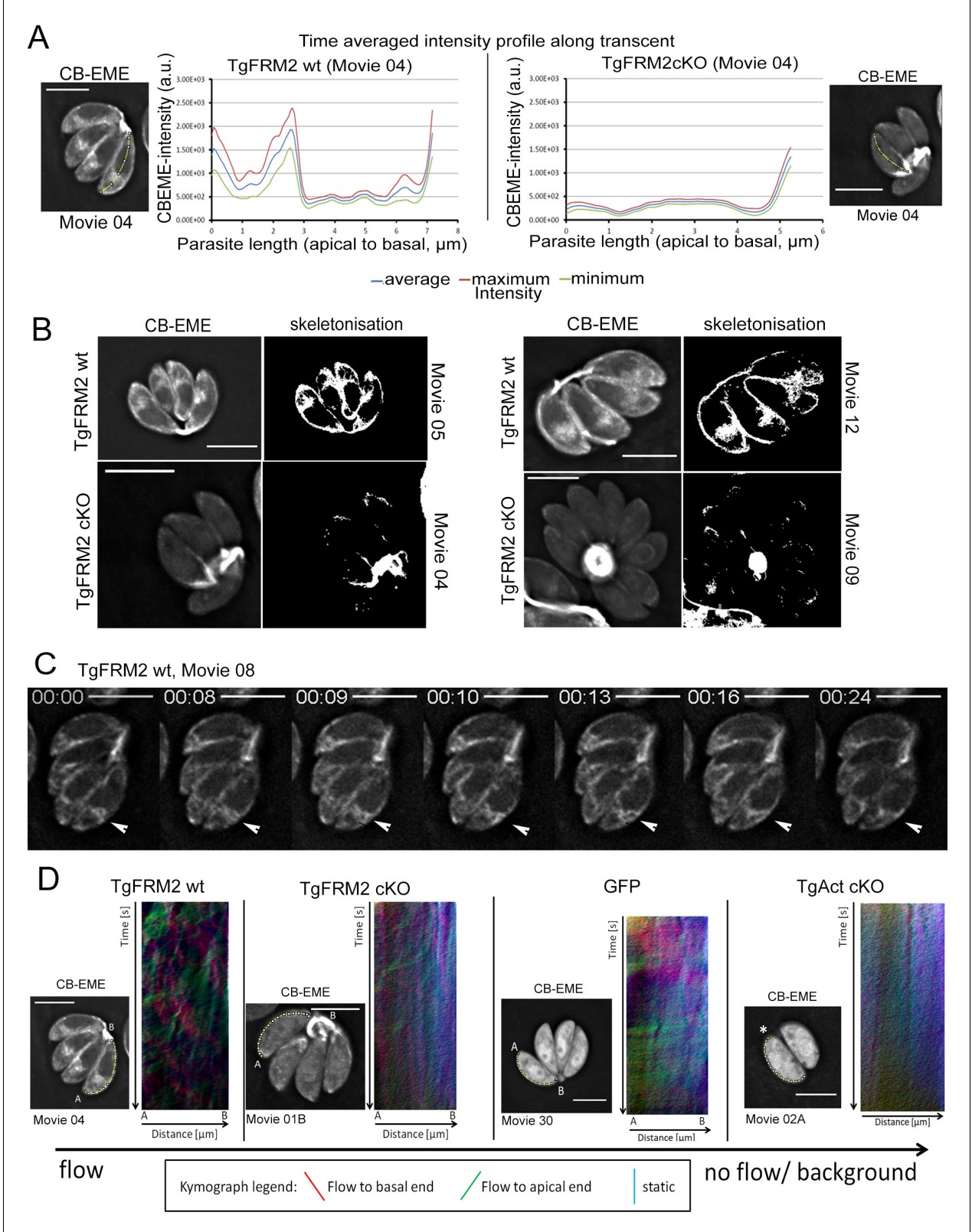

**Figure 8.** Intensity profiling and skeletonization of TgFRM2wt (control) and TgFRM2-Cas9cKO (TgFRM2cKO) parasites reveal major impact of TgFRM2 on intracellular F-actin distribution.  (A) Time-averaged local intensity profiling along the parasite middle axis (yellow line). Control TgFRM2 wt parasites show actin accumulation at the two poles and the region anterior to the nucleus (intracellular polymerization centre). Upon loss of TgFRM2, the intracellular actin polymerization centre is lost. Presence of actin at the apical pole appears reduced. (B) Skeletonisation analysis reveals actin

*Figure 8 continued on next page*

*Figure 8 continued*

accumulation at the periphery of intracellular parasites. Control parasites show actin at the two poles, the cytoplasmic intracellular polymerization centre and the periphery. TgFRM2 cKO mutants lack actin polymerisation within the region anterior to the nucleus and show less actin in the lateral space of the parasites. Actin still accumulates at the two poles. (C) Live imaging of TgFRM2 wt parasites illustrates the dynamic interaction between the cytoplasmic actin pool and the periphery of the parasite. Videos and skeletonisations are shown as images of collapsed t-stacks. At least 10 independent movies were produced and analysed for each condition. The figure shows representative images. Scale bars 5 μm. See also *Video 9*. (D) Kymograph analysis: Particle movement alongside the periphery was depicted via three colour-coded kymographs. Red tracks represent particles moving to the basal end, green tracks show particle flow to the apical end and blue depicts static particles. For TgFRM2 wt, kymographs show tracks (trajectories) of particle movement events to the apical and basal pole of the parasite. This suggests bi-directional flow of actin at the lateral parasite axis. Upon loss of TgAct, the kymographs appear more diffuse and depict only background particle flow (refer to *Figure 8—figure supplement 1*). While TgFRM2 cKO kymographs do not appear as diffuse as TgAct cKO kymographs, their overall flow events are less defined and more diffuse making them resemble RH-GFP kymographs. The yellow line represents the area of kymograph measurement. Particle movement was measured from the apical (A) to the basal pole (B). As polarity is difficult to define for TgAct cKO parasites, the start point of the flow measurement is indicated with an asterisk. Videos are depicted as collapsed t-stacks. At least 5 (actin) or 10 (others) independent movies were produced and analysed for each depicted condition. The figure shows representative kymographs. Conditional KO mutants represent Cas9cKO strains. WT parasites represent the non-induced TgFRM2-wt Cas9 strain. Scale bars are 5 μm. See also *Figure 8—figure supplement 1*, *Videos 9*, *10*, *11* and *12*.
DOI: https://doi.org/10.7554/eLife.49030.026

The following figure supplements are available for figure 8:

**Figure supplement 1.** Kymograph analysis reveals bi-directional peripheral actin flow in intracellular *Toxoplasma* parasites.
DOI: https://doi.org/10.7554/eLife.49030.027

**Figure supplement 2.** Time-averaged local intensity profiling and kymograph analysis for *Toxoplasma* parasites.
DOI: https://doi.org/10.7554/eLife.49030.028

perhaps not surprisingly, F-actin was still measurable and visible at the apical and the basal poles, as well as in the residual body of TgFRM2-cKO parasites. At these locations, actin polymerisation is probably mediated by the nucleation factors TgFRM1 and TgFRM3 which a recent study (*Tosetti et al., 2019*) localised to the apical tip and the residual body, respectively.

Tosetti and colleagues proposed peripheral actin flux towards the basal end of extracellular parasites (*Tosetti et al., 2019*). In a complementary approach, we applied kymograph analysis (*Mangeol et al., 2016*) for investigating actin flow at the periphery of intracellular parasites (*Figure 8D*, yellow tracks, *Figure 8—figure supplements 1* and *2*). Kymographs of wild-type parasites showed trajectories representing CB-EME particle movement towards the apical and basal poles (red signal represents forward displacement, while green signal represents backward displacement on kymographs, *Figure 8D*, *Figure 8—figure supplements 1* and *2*; *Videos 9*, *10*, *11* and *12*), indicating bi-directional actin flow all along the lateral parasite axis.

Time-averaged intensity profiling was also performed on *T. gondii* actin (TgAct) cKO and TgADF cKO parasites (*Figure 8—figure supplement 2A,C*, *Videos 10* and *11*) which displayed similar phenotypes described in previous reports (*Periz et al., 2017*): TgAct cKO parasites lacked any directed actin distribution, while disruption of TgADF let to actin accumulation, predominantly at the basal pole. As a control, GFP distribution was highly distinguishable from CB-EME distribution in parasites (*Figure 8—figure supplement 2E*, *Video 12*). In TgAct cKO parasites, no distinguishable tracks could be identified by kymograph analysis, when compared to background noise (*Figure 8—figure supplements 1* and *2B*, *Video 10*). TgADF cKO kymographs represent the previously described phenotype of strong static actin accumulation at the basal end of the parasite with no obvious F-actin dynamics at the periphery (*Figure 8—figure supplement 2D*, blue signal on kymographs, *Video 11*). Kymographs of TgFRM2 cKO parasites appear more diffuse, although some tracks could still be observed (*Figure 8D* and *Figure 8—figure supplement 1*, *Video 9*). This is most likely due to the abundance of actin in the periphery and perhaps due to the contribution of TgFRM1.

In summary, kymograph analysis demonstrates a striking difference in F-actin dynamics caused by disruption of TgAct, TgFRM2 or TgADF (*Del Rosario et al., 2019*). We conclude that loss of TgFRM2 causes a significant decrease of peripheral actin flow in intracellular parasites.

## Discussion

Due to the unconventional behaviour of apicomplexan actin, visualisation of actin filaments in *P. falciparum* was hindered by lack of reagents and F-actin sensors, which do not significantly interfere with

F-actin polymerisation and depolymerisation. Therefore, previous attempts to use established indicators from other eukaryotic systems, such as Life-Act, failed (*Tardieux, 2017*). In a recent study, it was shown that actin-binding nanobodies fused to epitope tags could be expressed in *Toxoplasma gondii,* allowing for the first time to analyse F-actin localisation and dynamics in living parasites (*Periz et al., 2017*). Another recent study showed that these nanobodies also bind to *P. falciparum* actin *in-vitro* (*Bookwalter et al., 2017*). Here we successfully adapted this technology to live *P. falciparum* parasites using two different epitope tags, the halo and the emerald tag. This allowed us to visualise F-actin throughout the asexual life-cycle of *P. falciparum* and in gametocytes (*Figure 1*) without causing any aberrant phenotypes, suggesting that this reagent, as in the case of *T. gondii* (*Periz et al., 2017*; *Del Rosario et al., 2019*) and all other eukaryotes tested so far (*Melak et al., 2017*), does not significantly interfere with F-actin dynamics. A recent study also used actin-binding chromobodies to analyse actin polymerisation centres in *T. gondii* and concluded that three Formins are responsible for actin dynamics (*Tosetti et al., 2019*). Importantly, validation of this reagent using either F-actin modulating drugs or a conditional mutant for PfACT1 led to expected results and phenotypes (*Figures 1* and *2*), demonstrating that F-actin dynamics are finely balanced in the parasite.

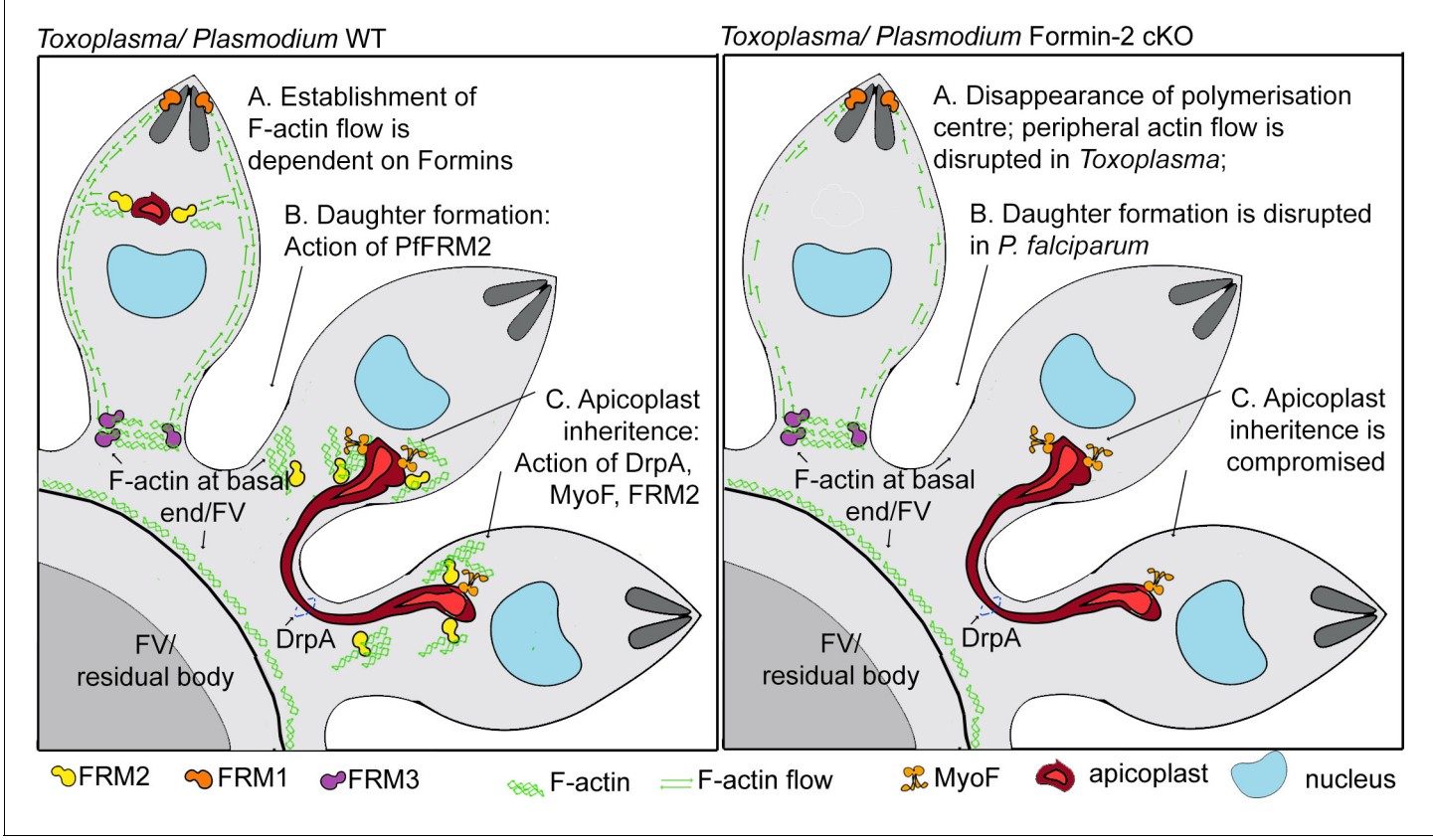

**Figure 9.** A model summarising Formin-dependent F-actin organisation, achieving apicoplast inheritance and daughter formation during intracellular replication of in *P. falciparum* and *T. gondii.* *Toxoplasma*/*P. falciparum* WT parasites: (**A**) Establishment of actin flow in intracellular parasites is dependent on Formin-2 (yellow) at the F-actin organising centre anterior to the nucleus, and may be further regulated by Formin-1 (orange) at the apical end and Formin3 (purple, Toxoplasma-specific) at the basal end. (**B**) Daughter cell formation in *P. falciparum* is controlled by Formin-2-dependent F-actin polymeriation at FV/basal end of the budding daughter cells. (**C**) Apicoplast inheritance in both *P. falciparum* and *T. gondii* occurs via the concerted action of DrpA-mediated scission, and MyoF-mediated pulling of apicoplasts along F-actin tracks created by Formin-2. *Toxoplasma*/*P. falciparum* Formin2 cKO parasites: (**A**) Polymerisation centres disappear with a complete lack of intracellular F-actin. In *Toxoplasma*, Formin-3 can still make extracellular F-actin connections between daughter cells. Establishment of bidirectional peripheral F-actin flow is disrupted. (**B**) Daughter cell formation in *P. falciparum* is compromised in Formin2 cKOs, possibly due to the lack of F-actin at the cytokinetic furrow. (**C**) Apicoplast inheritance in both *P. falciparum* and *T. gondii* is compromised due to the unavailability of tracks for movement of apicoplasts.
DOI: https://doi.org/10.7554/eLife.49030.033

Super-resolution imaging revealed a complex F-actin network in *P. falciparum* (*Figure 2*), similar to that observed in *T. gondii* (*Periz et al., 2017*) with extensive filaments around the FV – a location where the basal ends of the newly formed merozoites bud off during the end of schizogonic cytokinesis. We show that the primary nucleator of F-actin in intracellular *P. falciparum* is PfFRM2 and this protein also controls apicoplast inheritance and efficient cytokinesis (*Figures 3*, *4*, *5* and *6*). Importantly, our previous characterisation of a conditional mutant for PfACT1 highlighted three primary functions of actin during the asexual life cycle of the parasite (*Das et al., 2017*), which perfectly correlate to the localisation and dynamics found here using chromobody-expressing parasites. PfACT1 is essential for *P. falciparum* invasion into erythrocytes and we show for the first time the temporal and spatial dynamics of actin polymerisation by live microscopy prior to invasion. Despite a growing body of evidence suggesting the importance of calcium signalling and phosphorylation of IMC proteins by kinases such as calcium-dependent protein kinase-1 (CDPK1) and protein kinase A (PKA) (*Baker et al., 2017*; *Kumar et al., 2017*) during invasion, what triggers the polymerisation of actin is largely unknown. Our data suggest that early signalling events just after egress are a trigger for actin-polymerisation at the apical end. This is likely to be mediated by an apically resident nucleator of F-actin, a likely candidate being Formin-1 (*Baum et al., 2008b*), since PfFRM2 KO parasites could still polymerise actin at the apical end, as demonstrated in this study.

Therefore, the expression of chromobodies in *P. falciparum* allows us to phenotypically probe the state of the F-actin network *in-vivo* in a rapid and robust manner. F-actin can be clearly visualised during growth, in invading merozoites and in gametocytes – opening up many avenues for further research. It is conceivable that this novel tool could also be used to investigate other motile and developmental stages of the *Plasmodium* parasite.

Using actin-binding chromobodies combined with powerful reverse genetics made possible by the DiCre system (*Andenmatten et al., 2013*; *Collins et al., 2013a*) we show here that Formin-2 in both *Toxoplasma* and *Plasmodium* is required for the intracellular polymerisation of F-actin, a mechanism employed by the parasite for correct segregation of apicoplasts and cytokinesis. Using bioinformatic searches within alveolates, we found the presence of a PTEN-C2-like domain only in apicomplexan Formin-2 sequences (*Figure 3*). This domain has been demonstrated in rice to be responsible for Formin-2 targeting to chloroplast membranes (*Zhang et al., 2011*). It is therefore likely that the apicomplexan PTEN-C2-like domain is used for apicoplast recruitment of apicomplexan Formin-2.

We found that the function of Formin-2 is partially conserved in *T. gondii* and *P. falciparum*. In the case of *T. gondii,* however, the intra-vacuolar F-actin network is still formed (*Figure 7*), suggesting that *T. gondii* and potentially other coccidia have additional, compensatory mechanisms at their disposal to form this network, such as the presence of a Formin-3 (*Tosetti et al., 2019*). This network appears to be critical for material exchange, synchronised replication of parasites and host cell egress (*Periz et al., 2017*).

Furthermore, using kymograph analysis, we gained new detailed insights into actin distribution and dynamics in *Toxoplasma* tachyzoites. In addition to the previously reported actin polymerisation centres anterior to the nucleus and the residual body (*Periz et al., 2017*), we identified the apical and basal poles as sites of actin accumulation, and describe a bi-directional flow of actin along the cell periphery (*Figure 8*). Interestingly, our data suggest that these sites of actin accumulation interact with each other. We therefore propose that a steady flow of actin connects different sites of actin polymerisation, allowing for particle transport and exchange (*Figure 9*). Correspondingly, in a recent report it was elegantly demonstrated that forces set up by waves of actin polymerisation, along with actin comet formation, aided in segregation of yolk granules towards the vegetal pole of zebrafish oocytes (*Shamipour et al., 2019*).

Although *Tg*FRM2 is expendable for invasion and gliding (*Tosetti et al., 2019*), TgFRM2-mediated actin nucleation appears to be a major contributor to actin distribution in intracellular *Toxoplasma* parasites and in *P. falciparum*. Since *Tg*FRM2 cKO parasites still show actin polymerisation at the apical pole and in the residual body, and PfFRM2 cKOs can polymerise actin at the apical end, our findings support previous studies which reported non-overlapping functions for Formin-1,–2 (*Baum et al., 2008b*) and the coccidian specific Formin-3 (*Tosetti et al., 2019*). Our study supports the view that Formin-1 is active during gliding and invasion, while Formin-2 drives actin translocation and flow in intracellular apicomplexan parasites, achieving apicoplast inheritance in both

*Toxoplasma* and *Plasmodium*, and efficient daughter formation, observed only in *P. falciparum* (*Figure 9*).

In conclusion, we show here that chromobodies can be used to determine F-actin dynamics in apicomplexan parasites and will form the basis for functional *in-vivo* studies of other actin regulatory proteins found in apicomplexans.

# Materials and methods

## Key resources table

| Reagent type (species) or resource | Designation | Source or reference | Identifiers | Additional information |
|---|---|---|---|---|
| Gene (*Plasmodium falciparum*) | PfActin-1, PfACT1 PfFormin2, PfFRM2 | *Das et al. (2017)* *Baum et al., 2008b* | PF3D7_1246200 PF3D7_1219000 | |
| Gene (*Toxoplasma gondii*) | Tgformin2; Tgfrm2 | PMID: 22397711; *Tosetti et al., 2019* | TGME49_206580; TGGT1_206580 | |
| Gene (*Toxoplasma gondii*) | Tgactin1; Tgact1, TgAct | *Andenmatten et al., 2013*; PMID: 9227855; PMID: 8601316; *Whitelaw et al., 2017*; PMID: 26081631; *Egarter et al., 2014*; PMID: 21998582; PMID: 23921463; PMID: 22397711 | TGME49_209030; TGGT1_209030 | |
| Gene (*Toxoplasma gondii*) | Tgadf | PMID: 20042603; PMID: 21820516; PMID: 21346192 | TGME49_220400; TGGT1_220400 | |
| Transfected construct (*Plasmodium falciparum*) | pCB-EME and pCB-HALO | this paper | | actin-chromobody construct with emerald and HALO tags under the *P. falciparum* hsp86 promoter |
| Cell line (*Plasmodium falciparum*) | 1G5 DiCre strain | *Collins et al., 2013a* | | DiCre-expressing cell line |
| Cell line (*Plasmodium falciparum*) | B11 DiCre strain | *Perrin et al., 2018* | | DiCre-expressing cell line |
| Cell line (*Plasmodium falciparum*) | LoxPAct1 | *Das et al., 2017* | | The Actin-1 gene was floxed for DiCre-mediatedconditional excision |
| Cell line (*Plasmodium falciparum*) | LoxPPfAct1/CBEME | this paper | | The actin chromobody emerald construct pCB-EME was transfected on top of LoxPACT1 |
| Cell line (*Plasmodium falciparum*) | LoxPPfAct1/ CBHALO | this paper | | The actin chromobody HALO tagged construct pCB-HALO was transfected on top of LoxPACT1 |
| Cell line (*Plasmodium falciparum*) | LoxPPfFRM2-HA | this paper | | The Formin-2 gene was simultaneously floxed and tagged in the B11 DiCre strain |

*Continued on next page*

*Continued*

| Reagent type (species) or resource | Designation | Source or reference | Identifiers | Additional information |
|---|---|---|---|---|
| Cell line (*Plasmodium falciparum*) | LoxPPfFRM2-HA/ CBEME | this paper | | The actin chromobody construct pCB-EME was transfected on top of LoxPPfFRM2-HA |
| Cell line (*Homo sapiens*) | Human foreskin fibroblasts (HFF) | ATCC | ATCC SCRC-1041; RRID: CVCL_3285 | The cell line is commercially available at ATCC |
| Cell line (*Toxoplasma gondii*) | RHΔhxgprt | PMID: 8662859 | | |
| Cell line (*Toxoplasma gondii*) | RHΔku80 DiCre | *Hunt et al., 2019* | | Dr Moritz Treek (The Francis Crick Institute, London) |
| Cell line (*Toxoplasma gondii*) | TgFRM2-HA | this paper | | The Tgfrm2 gene was endogenously tagged with 3xHA at the c-terminus |
| Cell line (*Toxoplasma gondii*) | LoxPTgFRM2 (also referred to as LoxPTgFRM2-YFP-LoxP; referred to as TgFRM2 cKO or TgFRM2-YFP cKO upon excision of TgFRM2) | this paper | | The Tgfrm2 gene was floxed in RH Δ ku80 DiCre parasites and endogenously tagged at the c-terminus with YFP |
| Cell line (*Toxoplasma gondii*) | TgFRM2-Cas9wt (referred to as TgFRM2-wt when non-induced; referred to TgFRM2-Cas9cKO or TgFRM2-cKO when induced with Rapamycin) | this paper | | RH parasites expressing a conditional Cas9 system together with a gRNA targeting Tgfrm2 |
| Cell line (*Toxoplasma gondii*) | TgACT1-Cas9wt (referred to as TgACT1-wt when non-induced; referred to TgACT1-Cas9cKO or TgACT1-cKO when induced with Rapamycin) | this paper | | RH parasites expressing a conditional Cas9 system together with a gRNA targeting Tgact1 |
| Cell line (*Toxoplasma gondii*) | TgADF-Cas9wt (referred to as TgADF-wt when non-induced; referred to TgADF-Cas9cKO or TgADF-cKO when induced with Rapamycin) | this paper | | RH parasites expressing a conditional Cas9 system together with a gRNA targeting Tgadf |
| Cell line (*Toxoplasma gondii*) | RHΔhxgprt-GFP | other | | Dr Musa Hassan (Unibersity of Edinburgh, The Roslin Institute, Edinburgh); the gfp gene was randomly integrated into the parasite genome. |
| Antibody | mouse anti-actin | *Angrisano et al., 2012* | RRID: AB_2665920 | Polyclonal antibody raised against a parasite-specific polypeptide epitope. IFA dilution 1:500 |
| Antibody | Rat anti-haemagglutinin (HA) | Roche | cat# 1187431001 | Monoclonal antibody raised in rat |
| Antibody | rabbit anti-GFP | Abcam | cat #ab6556; RRI D:AB_305564 | Polyclonal antibody, IFA dilution 1: 500 |

*Continued*

| Reagent type (species) or resource | Designation | Source or reference | Identifiers | Additional information |
|---|---|---|---|---|
| Antibody | mouse anti-Atrx1 | (*DeRocher et al., 2008*) PMID:18586952 | | Polyclonal antibody, IFA dilution 1:500 |
| Antibody | rabbit anti-G2Trx | Biddau and Sheiner, unpublished. | | Polyclonal antibody, IFA dilution 1:500; Dr Lilach Sheiner (University of Glasgow, Institute of Infection, Immunity and Inflammation, Glasgow) |
| Antibody | rabbit anti-TOM40 | (*van Dooren et al., 2016*) PMID: 27458014 | | Polyclonal antibody, IFA 1:1000 |
| Antibody | rabbit anti-CPN60 (apicoplast) | (*Agrawal et al., 2009*) PMID: 19808683 | | Polyclonal antibody, reactive to Toxoplasma and *P. falciparum*. IFA dilution 1:2000 |
| Chemical compound, drug | Compound 2 | *Collins et al., 2013b* | | *P. falciparum* Protein Kinase G inhibitor |
| Software, algorithm | Ima ge J | *Schneider et al., 2012*; *Schindelin et al., 2012* | | |
| Software, algorithm | ImageJ plug-in 'KymographClear' | *Mangeol et al., 2016* | | |
| Software, algorithm | KymographDirect | *Mangeol et al., 2016* | | |
| Software, algorithm | Graphpad PRISM 7 ver 7.03 | GraphPad Software | | Commercial software for statistical analysis |

## Culture and transfection of *P. falciparum*

*P. falciparum* was cultured in O + human red blood cells from the Scottish National Blood Transfusion Service, at 37°C in RPMI 1640 with Albumax (Invitrogen) and schizonts were purified on a bed of 70% Percoll as described previously (*Blackman, 1994*). About 10 µg of plasmid was ethanol precipitated and resuspended in 10 µL sterile buffer TE (Qiagen). The Amaxa P3 primary cell 4D Nucleofector X Kit L (Lonza) was used for transfections. The input DNA was added to 100 µL P3 primary cell solution, mixed with 10–20 µL of packed synchronous mature schizonts and added to the cuvette, which was electroporated in a 4D-Nucleofector machine (Lonza) using the program FP158. The transfected schizonts were rapidly added to 2 mL of complete medium (RPMI with Albumax supplemented with glutamine) containing erythrocytes at a haematocrit of 15%, and left shaking in a shaking incubator at 37°C for 30 min. Finally the cultures were supplemented with 7 mL of complete RPMI medium to obtain a final haematocrit of 3% and incubated overnight at 37°C in a small angle-necked flask (Nunc). Parasites were selected by use of appropriate drug medium. The culture medium was subsequently exchanged every day for the next 4 days to remove cell debris which accumulates during electroporation and then twice a week until parasites were detected by Giemsa smear. Drug-resistant parasites were generally detectable in thin blood films 2–3 weeks post-transfection. After this, parasite stocks (at ~5% ring parasitaemia) were cryopreserved in liquid nitrogen. Lines were then cloned by limiting dilution using a simple plaque assay (*Thomas et al., 2016*).

## Cloning and expression of actin-chromobodies in *P. falciparum*

The CB-HALO and CB-EME plasmid consists of a sequence encoding actin chromobody from Chromotek followed downstream by an in frame sequence encoding Halo (Promega) or the emerald tag. CB-EME and CB-HALO sequences were amplified by PCR and cloned into the vector pB-map2gfpdd (Nicholas Brancucci, unpublished) via restriction sites NheI and HindIII to remove the map2gfpdd

sequence and put the CB-sequences under the *hsp86* promoter. The resulting plasmids pB-CBEME and pB-CBHALO were sequenced on both strands to confirm correct nucleotide sequences. These constructs were transfected as described into the loxPACT1 parasite clone B2 (*Das et al., 2017*) to obtain parasite lines LoxPPfACT1/CBEME, LoxPPfACT1/CBHALO and into the parental 1G5DiCre clone (*Collins et al., 2013a*) to obtain the line CBEME/1G5DiCre and CBHALO/1G5DiCre. Lines were selected with 2.5 µg/mL blasticidin. CB-EME expression was visible by fluorophore excitation/ emission in the green range and the HALO ligand was made visible by use of the ligand HALO-TMR at 1:40,000 with excitation/emission in the red range. Alternatively antibodies were used against the HALO tag to stain for CB-HALO.

### *P. falciparum* IFA

Thin blood films were made on glass slides and fixed in 4% paraformaldehyde in PBS for 20 min. The slides were then permeabilised with 0.1% Triton-X/PBS for 10 min, washed and blocked over-night in 4% BSA/PBS. Antigens were labelled with suitable primary and secondary antibodies in 4% BSA/PBS with 5 min PBS washes in between. Slides were finally air dried and mounted with DAPI-Fluormount-G (SouthernBiotech).

Staining of the RON4 junction in CB-EME expressing was performed by fixation and immunos-taining in solution as described previously (*Riglar et al., 2011*).

For image acquisition, z–stacks were collected using a UPLSAPO 100 × oil (1.40NA) objective on a Deltavision Core microscope (Image Solutions – Applied Precision, GE) attached to a CoolSNAP HQ2 CCD camera. Deconvolution was performed using SoftWoRx Suite 2.0 (Applied Precision, GE).

An Elyra S1 microscope with Superresolution Structured Illumination (SR-SIM) (Zeiss) was used for super-resolution imaging.

Colocalisation analysis was performed by using the Coloc two plugin in ImageJ and obtaining the Pearson´s R value for two defined channels.

### Time lapse microscopy of live *P. falciparum*

Video microscopy of *P. falciparum* schizont egress was performed as described previously (*Collins et al., 2013b*). Synchronised schizonts were Percoll purified and treated with 1 µM C2 in RPMI medium with Albumax (Gibco) for 4 hr. Microscopy chambers (internal volume ~80 µl) for observing live schizonts were built by adhering 22 × 64 mm borosilicate glass coverslips to micro-scope slides with strips of double-sided tape, leaving ~4 mm gaps at each end. C1 was washed off before video microscopy and the schizonts were immediately resuspended into warm (37˚C) RPMI (with Albumax) and introduced by capillary action into the pre-warmed chamber. The chamber was transferred to a temperature-controlled microscope stage at 37˚C on a Deltavision Core microscope (Image Solutions – Applied Precision, GE). Images were routinely collected at 5 s intervals, beginning 6 min 30 s after washing off C1, over a total of 30 min.

Other than during egress, CB-EME and CB-HALO expressing parasites were imaged at intervals of 1 s.

### Bioinformatics

Proteomes of interest (*Supplementary file 2*) were downloaded from the UniProt-KB website (www.uniprot.org). These were concatenated into a single proteome sequence dataset. All sequence iden-tifiers and annotations referred to are from UniProt Hidden Markov Models (PFAM profiles) PF02181.23 (FH2.hmm, Formin Homology 2 Domain) and PF10409.9 (PTEN_C2.hmm, C2 domain of PTEN tumour-suppressor protein) were downloaded from Pfam (*El-Gebali et al., 2018*). These pro-files were used with the HMMER package (HMMER 3.1b1 (May 2013); http://hmmer.org/) to search the proteome sequences (hmmsearch), and to align sequences of interest (hmmalign). The proteome sequence dataset was searched for FH2 domains (FH2.hmm) with hmmsearch, and sequences with regions scoring >28 bits recorded. These sequences were retrieved from the dataset, and subjected to alignment against the FH2.hmm. The profile conformant subsequences were extracted from the alignment and this sequence set subjected to alignment using: (1)hmmalign to FH2.hmm, (2) clustalw (*Thompson et al., 1994*) (3) muscle (*Edgar, 2004*) and (4) T_Coffee. These multiple sequence align-ments were combined and evaluated in T_coffee (*Keller et al., 2011*) using the -aln and -special_-mode evaluate options of T_coffee and the alignment edited to remove columns of average

quality <4 and occupancy <30% (T_coffee -other_pg seq_reformat option). A rooted neighbour-joining tree of Formin Homology type two domains (FH2) was constructed from this alignment using the SplitsTree program [version 1.14.8,*]. The proteome dataset was searched for the presence of PTEN_C2 conformant sequences. As only an inconsistent subset of sequences were found in both PTEN_C2 and FH2 selected sequences; one such subsequence (A0A1A7VGT3_PLAKH, residues 1096–1238) was used as the query of an iterative psi-blast [@], (E-value cutoff = 10) using the proteome data set as the database. The program converged after three iterations. The sequences flagged by psi-blast as having PTEN_C2-like sequence were compared with the sequences flagged by hmmsearch as having FH2 domains, and such sequences annotated on the phylogenetic tree.

## Creation of LoxPPfFRM2-HA and LoxPPfFRM2/CBEME strains

To obtain conditional truncation of the *pffrm2* gene we used silent *loxP* sites within a heterologous *P. falciparum* intron loxPint (*Jones et al., 2016*). We ordered from Geneart a ~ 800 bp targeting sequence followed by the LoxPint module in the context AATTGTAG-LoxPint-ATAGCTTT followed by a recodonised version of rest of the 3' region of the gene together with a C-terminal 3 HA tag. This ordered synthetic fragment was cloned into the pHH1-loxPMSP1 plasmid (*Das et al., 2015*) via restriction sites AflII and ClaI, replacing the msp1 sequence with *pffrm2*, giving rise to the plasmid pHH1-LoxPintFormin2 (*Figure 3G*). This was transfected into the DiCre expressing strain B11 (*Perrin et al., 2018*) and integrants selected by cycling on and off the drug WR99210 (Jacobus Pharmaceuticals, New Jersey, USA). The integrant line LoxPPfFRM2 was cloned by limiting dilution and two clones used for phenotypic characterisation. The strain LoxPPfFRM2/CBEME was created by transfecting the pB-CBEME plasmid into a LoxPPfFRM2-HA clone line and transfectants selected using the drug blasticidin (Sigma).

## Conditional truncation of *pfact1* and *pffrm2*

Various floxed parasite strains were synchronised by Percoll and sorbitol as previously described (*Collins et al., 2013b*). Briefly, schizonts were purified on a bed of 66% Percoll and allowed to reinvade into fresh erythrocytes for 1–2 hr. The remainder of the schizonts was removed by Percoll and the freshly invaded rings were subjected to 5% sorbitol for 7 min at 37°C to lyse any remaining schizonts. The tightly synchronised rings were divided into two flasks and pulse-treated for 4 hr at 37°C with 100 nM rapamycin or with 1% DMSO as control. The rings were then washed and returned to culture. Phenotypic analysis was performed primarily 44 hr post RAP-treatment unless stated otherwise.

## Culturing of *Toxoplasma* parasites and host cells

Human foreskin fibroblasts (HFFs) (RRID: CVCL_3285, ATCC) were grown on tissue culture-treated plastics and maintained in Dulbecco's modified Eagle's medium (DMEM) supplemented with 10% foetal bovine serum, 2 mM L-glutamine and 25 mg/mL gentamycin. Parasites were cultured on HFFs and maintained at 37°C and 5% $CO_2$. Cultured cells and parasites were regularly screened against mycoplasma contamination using the LookOut Mycoplasma detection kit (Sigma) and cured with Mycoplasma Removal Agent (Bio-Rad) if necessary.

## Microscopy for *Toxoplasma*

Widefield images were acquired in z-stacks of 2 μm increments and were collected using an Olympus UPLSAPO 100 × oil (1.40NA) objective on a Delta Vision Core microscope (AppliedPrecision, GE) attached to a CoolSNAP HQ2 CCD camera. Deconvolution was performed using SoftWoRx Suite 2.0 (AppliedPrecision, GE). Further image processing was performed using ImageJ software (*Schindelin et al., 2012*; *Schneider et al., 2012*).

Super-resolution microscopy (SR-SIM) was carried out using an ELYRA PS.1 microscope (Zeiss) as described previously (*Periz et al., 2017*). Images were acquired using a Plan Apochromat 63×, 1.4 NA oil immersion lens, recorded with a CoolSNAP HQ camera (Photometrics)using ZEN Black software (Zeiss) and subsequently analysed with ImageJ software (*Schindelin et al., 2012*; *Schneider et al., 2012*).

### *Toxoplasma* IFA

For immunofluorescence analysis, HFF monolayers infected with *Toxoplasma* parasites were grown on coverslips and fixed at the indicated time points in 4% paraformaldehyde for 20 min at RT. Afterwards coverslips were permeabilised in 0.2% Triton X–100 in 1 × PBS for 20 min, followed by blocking (3% BSA and 0.2% Triton X–100 in 1x PBS) for at least 30 min. The staining was performed using indicated combinations of primary antibodies (*Supplementary file 1*) for 1 hr and followed by secondary Alexa Fluor 488 or Alexa Fluor 594 conjugated antibodies (1 : 3000, Invitrogen – Molecular Probes) for another 45 min. Nuclei were stained with DAPI-Fluormount-G (SouthernBiotech).

## Generation of the TgFRM2-HA and loxPTgFRM2-YFP strains in RHδku80DiCre parasites

Guide RNAs targeting the upstream region of TgFRM2 and the C-terminal region were designed using EuPaGDT (*Peng and Tarleton, 2015*). These were cloned into a vector expressing a Cas9-YFP fusion as well as the specific gRNAs as previously described (*Curt-Varesano et al., 2016*). The designed gRNAs ACTTTTCATAGTATAGGAGG CGG and AATAGGGGTCTGTAGGTTAA GGG bind 989 bp upstream of the start codon and 12 bp upstream of the stop codon of TgFRM2 respectively. To introduce the upstream LoxP site, the LoxP sequence ATAACTTCGTATAGCATACATTATAC-GAAGTTAT flanked with respective 33 bp homology on each side was ordered as a 100 bp primer (ThermoFischer Scientific). The repair template for the C-terminal tag (HA or YFP) was generated by PCR using Q5 polymerase (New England Biolabs) from template plasmids with 50 bp of target-specific homology introduced via the primer. All tags are flanked by the same sequence, the upstream linker sequence GCTAAAATTGGAAGTGGAGGA encoding for the amino acid sequence AKIGSGG, the tag itself, a stop codon and the LoxP sequence. The YFP tag is superfolder YFP 2, and was subcloned from pSYFP2-C1 (gift from Dorus Gadella (Addgene plasmid # 22878; http://n2t.net/addgene:22878; RRID:Addgene_22878) (*Kremers et al., 2006*). All C-terminal repair templates were pooled, purified using a PCR purification Kit (Blirt). Together with 10 µg Cas9 vector encoding the respective gRNA, 1 × 107 of freshly released RHΔku80DiCre tachyzoites (an improved version created by Dr Moritz Treeck *Hunt et al., 2019* from the original; *Andenmatten et al., 2013*) were transfected using 4D AMAXA electroporation. 24 hr after transfection, parasites were mechanically released, filtered and sorted for transient YFP expression into 96 well plates using a FACS sorter (FACSARIA III, BD Biosciences). Individual plaques were screened by PCR and the C-terminus of TgFRM2 was sequenced (Eurofins Genomics). Into a clone with TgFRM2-YFP-LoxP, the upstream LoxP was introduced as described. Screening for upstream LoxP integration was performed by PCR with a primer binding at the junction of gRNA binding sequence and LoxP site. Using a different set of primers, the complete upstream LoxP site was amplified via PCR and verified by sequencing. Two distinct clones were obtained for LoxPTgFRM2-YFP-LoxP (clone A and B) and used for phenotypic characterisation.

## Induction of the conditional DiCre TgFRM2 KO parasites

To obtain TgFRM2-YFP KO parasites, the loxPTgFRM2-YFP-LoxP parental line was grown in 50 nM rapamycin containing media as described above until fixing. In IFA, TgFRM2-YFP KO parasites were always compared to a control population of untreated loxPTgFRM2-YFP-LoxP.

## Induction of the conditional CRISPR/Cas9 cKO mutants

A conditional CRSIPR/Cas9 system was used to disrupt the genes actin1, adf and formin2 (Stortz, Grech et al. in preparation). Conditional CRISPR/Cas9 knock-out (cKO) mutants for actin1, adf and formin2 were obtained by adding 50 nM rapamycin to the parental lines expressing the conditional CRISPR/Cas9 system and a gene-specifc gRNA. Parasites were incubated with rapamycin for 1 hr at 37°C and 5% $CO_2$ and, subsequently, cultured as described previously. For the CRSIPR/Cas9 actin1-cKO mutants, the culture medium was replaced by DMEM complete supplemented with 2.5% dextran sulphate after 24 hr to inhibit re-invasion of wild-type parasites. Disruption of the target genes was confirmed by sequencing (Eurofins Genomics).

### Transient transfection of CB-EME into *Toxoplasma* parasites

To have parasites transiently expressing CB-EME, $1 \times 10^7$ of freshly released TgFRM2-HA or loxPTgFRM2-YFP parasites were transfected with 20 µg DNA by AMAXA electroporation. Subsequently, parasites were grown on HFFs as described above and fixed with 4% paraformaldehyde after 48 hr or 72 hr.

### Time-lapse video microscopy for *Toxoplasma*

Conditional CRSISPR/Cas9 strains were grown on fresh HFF cells for 72 hr as described above. Subsequently, parasites were mechanically lysed and inoculated on glass bottom dishes (MaTek) for another 24 hr. RH-GFP parasites were inoculated on glass bottom dishes (MaTek) for 24 hr. Prior to live microscopy, the DMEM complete culturing media was replaced with FluoroBrite DMEM media supplemented with 10% foetal bovine serum, 2 mM L-glutamine and 25 mg/mL gentamycin. The dish was then transferred to the DV Core microscope (AppliedPrecision, GE) and maintained under standard culturing conditions (37°C, 5% $CO_2$). Images were taken using a 100x oil objective lens. Deconvolution was performed using SoftWoRx Suite 2.0 (Applied Precision, GE). Videos were processed using ImageJ (*Schneider et al., 2012*; *Schindelin et al., 2012*).

### Generation of colour-coded kymographs for particle dynamics analysis and time-averaged local intensity profiles

Colour-coded kymographs were generated by applying the ImageJ plugin 'KymographClear' as described previously (*Mangeol et al., 2016*). In short, we used this application to define a track on a maximum intensity image that was calculated from an image sequence. A kymograph was then generated depicting particle movement alongside the chosen track. Fourier filtering done by the plugin enables the distinction between forward-moving (red), backward-moving (green) and static (blue) particles in the kymograph.

Kymograph data were exported to the stand-alone software 'KymographDirect' to generate time-averaged local intensity profiles (*Mangeol et al., 2016*). Intensity profiles depict Cb-Emerald intensity along the measured axis over the entire duration of the movie. Background corrections were performed for all imported kymographs.

### Skeletonisation of videos obtained from live microscopy

Image sequences were skeletonised with the ImageJ plugin 'Skeleton' (*Schindelin et al., 2012*). Prior to skeletonization, thresholding was performed on the movie stacks to create binary images defining signal and no signal. These binary images were then processed by the skeletonization plugin, converting the signal into pixels that can be followed through the time-lapse. Skeletonized images in this study represent collapsed t-stacks.

## Acknowledgements

We thank Prof. Mike Blackman for the kind gift of the PKG inhibitor Compound two and the B11 DiCre strain. We thank Dr. Jake Baum for the PfACT1 and RON4 antibodies, Dr. Julian Rayner for the MTIP and GAP45 antibodies, Dr Lilach Sheiner for the CPN60 and G2Trx antibodies, Prof Peter Bradley for the Atrx1 antibody and Prof. GK Jarori for the enolase antibodies.

## Additional information

### Funding

| Funder | Grant reference number | Author |
|---|---|---|
| H2020 Excellent Science | ERC-2012-StG 309255-EndoTox | Markus Meissner |
| Wellcome | Wellcome Senior Fellowship 103875/Z/14/Z | Markus Meissner |

| Horizon 2020 Framework Programme | LMU Fellowship H2020-MSCA-COFUND-2016-754388 | Sujaan Das |
| Wellcome | 085349 - Core funding for the WCMP | Johannes Felix Stortz<br>Mario Del Rosario<br>Jonathan M Wilkes<br>Markus Meissner<br>Sujaan Das |
| National Secretariat for Higher Education, Sciences, Technology and Innovation of Ecuador (SENESCYT) PhD scholarship | IFTH-GBE-2015-0475-M | Mario Del Rosario |

The funders had no role in study design, data collection and interpretation, or the decision to submit the work for publication.

## Author contributions

Johannes Felix Stortz, Formal analysis, Investigation, Methodology, Writing—original draft; Mario Del Rosario, Formal analysis, Validation, Methodology; Mirko Singer, Investigation, Methodology; Jonathan M Wilkes, Formal analysis, Investigation, Methodology; Markus Meissner, Conceptualization, Resources, Supervision, Writing—original draft, Project administration, Writing—review and editing; Sujaan Das, Conceptualization, Formal analysis, Supervision, Funding acquisition, Investigation, Methodology, Writing—original draft, Project administration, Writing—review and editing

## Author ORCIDs

Johannes Felix Stortz (iD) https://orcid.org/0000-0002-5928-1850
Mario Del Rosario (iD) https://orcid.org/0000-0002-0430-1463
Mirko Singer (iD) https://orcid.org/0000-0002-5757-2750
Markus Meissner (iD) https://orcid.org/0000-0002-4816-5221
Sujaan Das (iD) https://orcid.org/0000-0002-6466-4258

## Decision letter and Author response

Decision letter https://doi.org/10.7554/eLife.49030.039
Author response https://doi.org/10.7554/eLife.49030.040

# Additional files

## Supplementary files

• Supplementary file 1. Antibodies used in this study.
DOI: https://doi.org/10.7554/eLife.49030.034

• Supplementary file 2. Proteomes of interest.
DOI: https://doi.org/10.7554/eLife.49030.035

• Transparent reporting form
DOI: https://doi.org/10.7554/eLife.49030.036

## Data availability

All data generated or analysed during this study are included in the manuscript and supporting files. Proteomes of interest (Supplementary File 2) were downloaded from the UniProt-KB website (www.uniprot.org).

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
