## [Decision Letter]

[Editors’ note: a previous version of this study was rejected after peer review, but the authors submitted for reconsideration. The first decision letter after peer review is shown below.]

Thank you for submitting your work entitled "Formin-2 drives polymerisation of actin filaments enabling segregation of apicoplasts in *P. falciparum* and *T. gondii*" for consideration by *eLife*. Your article has been reviewed a Senior Editor, a Reviewing Editor, and two reviewers. The following individual involved in the review of your submission has agreed to reveal their identity: Isabelle Tardieux (Reviewer #1).

Our decision has been reached after consultation between the reviewers. Based on these discussions and the individual reviews below, we regret to inform you that your work will not be considered further for publication in *eLife*.

This work builds on a previous study aimed at visualizing the actin cytoskeleton in Apicomplexa using chromobodies. As you will see from the reviewer reports appended below, both reviewers recognized the importance of the topic. The reviewers discussed your work in detail, and the consensus was that although the studied problem is challenging and the described work has some potential, in the current form the conclusions are not sufficiently supported by data, such as complementary approaches to provide sound evidence for the rather strong statements on the identity of the studied structures. Another strong point of criticism, on which both reviewers agreed, was the lack of quantitation throughout the paper. Since these concerns cannot be easily addressed within a short time frame, we advise you to seek publication elsewhere.

*Reviewer #1:*

The manuscript entitled "Formin-2 drives polymerisation of actin filaments enabling segregation of apicoplasts in *P. falciparum* and *T. gondii*" by JF Stortz et al., reports on the use of actin chromobodies to detect F-actin structures in the two Apicomplexa *Plasmodium falciparum* and *Toxoplasma gondii* parasites, and on the involvement of these structures during parasite intracellular development. While the authors have already obtained *T. gondii* lines expressing the actin chromobodies (Periz et al., 2017), this is newly achieved for *Plasmodium falciparum*. Using this tool to characterize putative F actin structures, they then investigated how intracellular dividing parasites (*P. falciparum* merozoite to schizont in the red blood cells) or *T. gondii* tachyzoites in nucleated mammalian cells) are affected when the actin1 gene is conditionally silenced using the DiCre recombinase/ lox system. They confirmed for *P. falciparum* asexual erythrocytic stages that loss of actin impacts proper segregation of the apicoplast organelle, a finding already reported by the authors for *T. gondii* tachyzoites (Whitelaw et al., 2017). The manuscript is then centered on the key role of actin dynamics to promote apicoplast segregation. Given this observation the authors established an association between pools of actin chromobody-bound actin and apicoplasts using confocal and super resolution imaging. To start addressing how actin dynamics could control apicoplast, they raised the hypothesis that the actin nucleator Formin (here Formin-2) would be a relevant candidate. They showed using both epitope tagging at the endogenous locus and conditional silencing, that (i) main pools of Formin-2 are indeed juxtaposed to apicoplasts and (ii) loss of formin induces the breakdown of the F-actin structure and eventually apicoplast segregation defect.

It is noteworthy that this study relies on a combination of recently introduced but still challenging molecular genetics in the Apicomplexa field. The authors have engineered several lines of interest in both *T. gondii* and *P. falciparum* and this represents a valuable achievement. In addition, the view that Formin-2 could play a key role in controlling actin at specific locations during cell cycle is also quite interesting and rather convincingly documented. However, in my opinion the work should be strengthened by few additional datasets and by careful reevaluation of several statements prior to publication.

First and main point is the need to document in better detail how F-actin pools form/disorganize in space and time during the apicoplast segregation process in wild type parasites. Real time and static SR imaging should allow capturing the kinetics of the segregation process as well as spatial details (Z stacks thus 3D reconstitution will reveal them). Then, the use of conditional gene silencing will take all its power to support/contradict the scheme.

In Figure 2: it is very important that the authors do not refer to filopodia when observing fluctuations at the second time scale in the actin chromobody signals during the early stages (ring/trophozoite which should be indicated in the figures). If they want to characterize filopodia occurrence, they need to provide a marker for membrane and to analyze whether the parasite actin protrusions coincide with membrane projections.

*Reviewer #2:*

This work continues some previous studies that showed in *Toxoplasma* that one could get some visualisation of the actin biology by expressing fluorescent probes called chromobodies. Here the authors use this same technology to reanalyse the distribution of actin in both *Plasmodium* and *Toxoplasma*. The authors show some potentially interesting results – mainly built around interpretation of images relating to the apicoplast and implicating Formin-2 in the polymerisation of actin filaments that are known from previous work to be involved in apicoplast division.

At the heart of the work are some statements that define the fluorescent dots or areas of brightness as actin filamentous structures or apicoplasts etc. I absolutely recognise the difficulty of these visualisation experiments in terms of getting an expressed probe to be at a meaningful level of expression and then imaging its distribution and interpreting the results. I recognise that one is maybe not likely to expect to see huge filamentous actin cables such as one can see in even small cells such as yeast.

However, the essence of the matter is that I simply do not see that the authors have provided either direct evidence or complementary approaches (biochemistry or electron microscopy) to provide sound evidence for their rather strong statements of identity of the structure. I am sorry to say that I think this is an example of pushing one technology too far and not being realistic about how much one can rely on the interpretation of an image above background to be significant.

Starting early in the paper on Figure 1 there is a disconnect between the statements made and the images shown. Figure 1B contains images of CB-EME (and CB-HALO) on which the whole paper rests. The problem is that it is extremely difficult if not impossible to discern a filamentous structure in these images over background. There are a few patches of greater fluorescence than background but what allows one to say that these are filamentous? Second Cytochalasin B is supposed to depolymerise these F-actin filaments but there is still a lot of fluorescence in this cell. Is this merely CB-EME fluorescence not bound to actin (G-actin) or is it actin / CB-EME complex fluorescence? The authors want to use this to define that the reagent is specifically recognising actin filaments but I am unconvinced by these images. All of this can be bolstered (and surely must be?) by expressing the *Plasmodium* actin in vitro and doing some biochemistry. Does CB-EME etc. bind in vitro only to polymerised F-actin or does it also form a complex with G-actin? These patches could easily be localised unpolymerized G-actin.

Given the large background seen after Cytochalasin B as well as in the normal cells one needs some quantitative statements about comparative camera settings for image capture comparisons.

Figure 1C. The anti-actin antibody seems to give different staining and is deemed (subsection “Chromobodies label F-actin structures in *P. falciparum* asexual stages and in gametocytes”) to recognise both G and F actin. There is no biochemistry on the antibody – (blot of whole cell extracts, IP, etc.) so specificity not known. I don't feel confident about this conclusion when I compare the images.

Subsection “Chromobodies label F-actin structures in *P. falciparum* asexual stages and in gametocytes”. If chromobodies can bind to PfACT1 and actin2 (where is biochemical evidence?) then why conclude that the filaments could be built from both proteins? Why not of individual ones or a mix?

Subsection “Highly dynamic F-actin-rich filopodia-like structures extend outward from the periphery of growing parasites”. Structures are deemed to be highly dynamic and changing. This is very unscientific language given the data. What does it mean? Do the structures disappear and reappear? Do they move? Increase in size and decrease? Also, whatever they do can numbers not be given – time, number of events observed, number of types of events, etc.

Subsection “Chromobodies label F-actin structures in *P. falciparum* asexual stages and in gametocytes”. "passes through the whole cell" – unfortunate use of English.

Subsection “Chromobody labelled F-actin structures disappear upon disruption of PfACT1”. Figure 3F the authors speak about the close association of apicoplast with F actin. Actually, all one can see on this image is a large collection of dots. Why are there so many of different types if there should be only one apicoplast per cell?

Moreover, there is actually little coincidence by my eye of these dots. Surely the authors can quantify their statements by morphometric analysis. My conclusion is bolstered by the fact that there is very little yellow (overlapping green and red) on the merged image. Hence, their data don't justify their interpretation.

Subsection “Chromobody labelled F-actin structures disappear upon disruption of PfACT1”. When disrupted they say that there is a consequential apicoplast segregation defect. I don't see this. In Figure 3F the apicoplast dots are smaller – why would this be if a segregation phenotype?

Again, there are no numbers in the manuscript, no sizes, counts or morphometric analyses of juxtapositions.

Subsection “DiCre-mediated conditional disruption of Formin-2 causes a defect in apicoplast segregation in *P. falciparum”.* The authors speak about collapsed or morphologically aberrant apicoplasts? What is a collapsed apicoplast? No correlative EM is used. They speak about collapsed/intermediate/normal apicoplasts without any definition. All one can see are a series of dots. There is nothing to provide a definition that a particular dot represents a particular "type" of apicoplast. Again, no numbers, counts etc. No statistics. What is a collapsed apicoplast and what does it look like at a sensible resolution – i.e. use electron microscopy.

One can continue with this type of analysis in further experiments.

Actin filaments are known from previous work to be involved in apicoplast division and this paper seeks to implicate Formin in that process. This is a not unexpected factor to be involved. If the paper was to merit publication in such a journal it would need a stronger functional message based on more quantitative microscopy, some statistical analysis and some biochemistry. I am afraid this is a microscopy technique that seems to have overtaken the problem in a non-critical manner.

[Editors’ note: what now follows is the decision letter after the authors submitted for further consideration.]

Thank you for submitting your article "Formin-2 drives polymerisation of actin filaments enabling segregation of apicoplasts and cytokinesis in *P. falciparum*" for consideration by *eLife*. Your article has been reviewed by Anna Akhmanova as the Senior and Reviewing Editor, and three reviewers. The following individual involved in the review of your submission has agreed to reveal their identity: Tobias Spielmann (Reviewer #1).

The reviewers have discussed the reviews with one another and the Reviewing Editor has drafted this decision to help you prepare a revised submission.

This paper employs chromobodies to document the localization and dynamics of the actin cytoskeleton in *Plasmodium* and *Toxoplasma* and provides insight into the function of formin in regulating the formation of different actin structures in these parasites. Overall, the reviewers were positive about the quality of the data and the strength of the conclusions. Therefore, the reviewers agreed that it was not necessary to perform additional experiments, but that the paper would benefit from improving the writing and the presentation of the data. We include below the individual reviews and hope that you will finding them helpful when revising your paper.

*Reviewer #1:*

Filamentous actin has been notoriously difficult to detect in malaria parasites. In this paper the authors use chromobodies to visualise F-actin in live malaria parasites. They validate the observed structures to be F-actin using CytD and Jaspla and in a conditional actin KO. Next they generated a conditional Formin-2 KO and used the actin chromobody to assess the impact on actin. This permitted them to nicely link phenotypes of the Formin-2 KO with changes in F-actin pools in the cell. This is most convincing with the apicoplast which shows a segregation phenotype (fitting with the phenotype of the conditional actin KO that the authors previously showed in Das et al., 2017). The manuscript also shows that knocking out Formin-2 leads to loss of the F-actin network around the food vacuole. This network is present in late stage malaria parasites and likely is equivalent of the intravacuolar actin network in *Toxoplasma* parasites. The dependence of this network on Formin-2 is in contrast to the situation in *T. gondii*, where this network is dependent on a third formin that is not present in malaria parasites. These findings are shown in the last part of the manuscript where the authors analyse F-actin in a conditional Formin-2 KO in *Toxoplasma*. In part this overlaps with data from a recent report in *eLife* (Tosetti et al., 2019) that also used the actin chromobody of the authors of the present manuscript. Interestingly, the work here indicates bi-directional F-actin flows along the parasite periphery whereas the previous report by Tosetti et al., indicated unidirectional flow toward the basal end.

Together with other work, this paper leaves us with a good picture which formin drives different actin-mediated processes in plasmodium and toxoplasma parasites. It convincingly shows that Formin-2 plays a central role for most actin-dependent processes in malaria blood stages (including the functions of Formin3 in *T. gondii*) but not the invasion functions which are controlled by Formin-1. A highlight of the paper is the visualisation of F-actin in living malaria parasites, which is also demonstrated in a number of nice time-lapse movies. The actin chromobody will be an excellent tool for further functional studies in this parasite. Overall this is technically very challenging work that includes multiple conditional knock outs and is experimentally sound.

The only larger concern I have is the cytokinesis phenotype in the Formin-2 KO (but this may be solved by simply changing the text to allow other options). Can the authors exclude that other phenotypes in later trophozoites or early schizonts contribute to the phenotype? Many different cellular defects could lead to the reduced number of nuclei in Figure 6A. In fact, this figure most likely indicates a general developmental delay rather than a specific phenotype. Do the 'slower' cells with less than 5 nuclei in Figure 6A still make it to egress? Or do they have other defects that make them arrest earlier? If they do not reach the late schizont, it may not only be a cell division phenotype. Was Figure 6E done with purified late stages or with all parasites on/off RAP?

The subsequent experiments (Figure 6B etc.) are then done with purified late stage parasites which consists of the subfraction that makes it to that stage (maybe those parasites where Formin-2 levels dropped more slowly after excision of the gene?). These experiments are important, as they show that the merozoites produced from this subfraction are still invasive and show that the Formin-1 dependent functions are not affected. The experiments are also consistent with the author's conclusion that this signifies a role of Formin-2 in 'coordinate cytokinesis'. However, it does not exclude that Formin-2 may have earlier function in schizogony. Unless all cells on RAP eventually make it to that stage, this may be just one of the phenotypes. This should be discussed.

*Reviewer #2:*

This is in principle a well performed study investigating actin dynamics in *P. falciparum* blood stages and *T. gondii* using a fluorescent antibody against actin filaments and a focus on the actin polymerizing protein Formin-2. The authors confirm and expand on recent work published in *eLife* and elsewhere introducing the chromobody, showing the importance of three different formins on actin polymerization in *T. gondii* as well as the finding that actin is important for apicoplast division. There is quite some overlap on the Formin-2 part with the recent paper from the Soldati group, but this should be covered by the scoop protection mechanism offered by *eLife*. The authors do a good job dissecting the differences in the roles of actin in *Plasmodium* and *Toxoplasma*.

This paper improves on the previously published *eLife* article of the same lab by examining for the first time actin dynamics with a chromobody in *P. falciparum* and by providing enhanced image analysis, which led to the discovery of a bidirectional cytoplasmic flow of actin filaments.

I don't have suggestions for additional experiments but maybe the authors can reflect on/discuss the following points:

It would improve reading experience if the movies would be somehow fused into fewer total movies, if possible – e.g. Video 1, Video 2, Video 3 and Video 4 could be put back to back of each other into a single file.

Can the structures visualized by the chromobody be bleached to investigate their turnover?

Does the bi-directional flow observed here compare with the recently reported actin flow in oocysts by the Heisenberg lab (2019)? How does this fit to the data recently published in *eLife*, where actin flows only from front of *T. gondii* to their back?

*Reviewer #3:*

This manuscript describes a technological breakthrough in the study of *P. falciparum*, an organism of great clinical significance, as well as reveals novel details on this organism's actin cytoskeleton, and draws parallels and distinctions for the roles Formin-2 in this organism and the distantly related *T. gondii*. This appears to be the second round of reviews for this manuscript, although this is my first viewing of the manuscript. Overall, I am satisfied with the rigor shown, and with the major conclusions that the authors draw. In particular, I am comfortable with the authors' demonstration that the chromobodies (CB's) are indeed faithfully decorating actin-based structures, and based on their sensitivity to CytoD, agree that these structures are likely F-actin-based. As outlined below, my main concerns regarding this work are the specific wordings used at some places in the manuscript regarding "dynamics", some omissions or gaps in the figure descriptions that cause confusion, and some missing information regarding the phylogenetic analysis.

1) The authors frequently use the term "dynamics" when describing actin-based structures. Based on their data, I would agree that many of the structures they view are dynamic in the sense that are mobile (e.g. subsection “F-actin in gametocytes”). However, "dynamic" with regards to actin also often refers to the degree to which the actin filaments undergo turnover (rounds of polymerization and depolymerization). The authors should review their text and edit where it needs to be made clear which type of dynamics they are discussing. Related to this, the authors claim to show that "islands of F-actin are stabilized upon jasplakinolide treatment (+JAS) (Figure 1C)", but this is documented only by a still image with large fluorescent bodies. The authors cite Video 3, but there is no +JAS movie on that video, only control and +CytoD. I agree that the authors could interpret their result as being consistent with stabilization of F-actin, but without showing that there is reduced turnover of actin in these structures, they should hedge on their wording some.

2) I also strongly recommend the authors supplement some of their figures/figure legends to help the reader better interpret the data. In particular:

For Figure 2Bii, the legend explains the significance of the bottom panels, but not the top four panels. Are they different time points?

For Figure 3G, the figure legend should provide some more information to help the reader understand the model. That is, "syn", "loxPint", and "WR" should be defined. It is a very elegant construct but requires a lot of flipping back and forth between the legend, text, and methods to deconstruct what is going on.

For Figure 3H, it currently gives the impression that Integrant 1 and Integrant 2 are being analyzed, when in fact it is transformant 1 and transformant 2 that are being analyzed, using a PCR reaction that probes for integration or the endogenous locus. I would suggest clarifying the labels, such as replacing "Integrant" and "Endogen." with "PCR(Int)" and "PCR(End)" (with clarifications in the legend), and add the label "Transformant:" to the left of "1 2 1 2" above the gels.

For Figure 4C, the x-axis is confusing to me. Perhaps this is because I do not study *P. falciparum*, but that will be true of a large number of readers.

For Figure 4D, what are the units for the y-axis? Is it an arbitrary value? A ratio of HA signal to enolase?

For Figure 7—figure supplement 1, reproduce the map shown in Figure 7A to help the reader to interpret the PCR results without needing to flip between two figures.

For Videos 9 – 11, multiple strains are shown simultaneously, but I could not observe any labels that identified which panel in a video corresponded to which strain. This was true when I viewed the movies directly from the website, and when I downloaded the mp4's.

3) Related to the phylogenetic analysis, Table T2, which lists the genomes that were searched for formins for the phylogenetic analysis, appears to be absent. Also, Figure 3B presents a single tree, but the Materials and methods section refers to "trees" (plural) that were constructed from "this alignment (or subsets of it)". The authors should clarify this. It would also be appropriate to provide the aligned sequences as a supplementary text file, if possible.

---

## [Author Response]

[Editors’ note: the author responses to the first round of peer review follow.]

Reviewer #1:[…] In addition, the view that Formin-2 could play a key role in controlling actin at specific locations during cell cycle is also quite interesting and rather convincingly documented. However, in my opinion the work should be strengthened by few additional datasets and by careful reevaluation of several statements prior to publication.

We thank the reviewer for their positive statements and for the subsequent feedback they have provided.

First and main point is the need to document in better detail how F-actin pools form/disorganize in space and time during the apicoplast segregation process in wild type parasites. Real time and static SR imaging should allow capturing the kinetics of the segregation process as well as spatial details (Z stacks thus 3D reconstitution will reveal them). Then, the use of conditional gene silencing will take all its power to support/contradict the scheme.

We thank the reviewer for this suggestion and agree that measurement of F-actin dynamics is very useful to document the influence of Formin on F-actin dynamics and function. In our previous submission we were not yet able to do so, since novel imaging tools were required. Therefore, we adapted kymograph analysis to measure F-actin flow in apicomplexans. We have now provided in Figure 1 and Video 1, Video 2, Video 3, Video 4 representations of how F-actin organizes and disorganizes in real time. Furthermore, to quantifiably describe the spatiotemporal dynamics of F-actin we have performed time averaged intensity measurements along defined transects and kymograph analysis (Figure 2, Figure 8 and Figure 2—figure supplement 2, Figure 8—figure supplement 1, Figure 8—figure supplement 2) and reliably shown the presence of defined pools of actin and actin flow in specific subcellular locations.

In Figure 2: it is very important that the authors do not refer to filopodia when observing fluctuations at the second time scale in the actin chromobody signals during the early stages (ring/trophozoite which should be indicated in the figures). If they want to characterize filopodia occurrence, they need to provide a marker for membrane and to analyze whether the parasite actin protrusions coincide with membrane projections.We no longer use the term ‘filopodia’ and describe them simply as actin accumulations.Reviewer #2:

*[…] Starting early in the paper on Figure 1 there is a disconnect between the statements made and the images shown. Figure 1B contains images of CB-EME (and CB-HALO) on which the whole paper rests. The problem is that it is extremely difficult if not impossible to discern a filamentous structure in these images over background. There are a few patches of greater fluorescence than background but what allows one to say that these are filamentous? Second Cytochalasin B is supposed to depolymerise these F-actin filaments but there is still a lot of fluorescence in this cell. Is this merely CB-EME fluorescence not bound to actin (G-actin) or is it actin / CB-EME complex fluorescence? The authors want to use this to define that the reagent is specifically recognising actin filaments but I am unconvinced by these images. All of this can be bolstered (and surely must be?) by expressing the Plasmodium actin* in vitro *and doing some biochemistry. Does CB-EME etc. bind* in vitro *only to polymerised F-actin or does it also form a complex with G-actin? These patches could easily be localised unpolymerized G-actin.*

We provide several lines of evidence that CB recognises specifically F-actin. First, we use CD and JAS and demonstrate polymerisation and depolymerisation respectively. It can clearly be seen in Figure 1 and Video 3 that all filamentous structures disappear under CD. Second, we also use a genetic disruption of actin-1 and show that filaments disappear. We have now developed and provided a quantitative method (kymograph analysis and local time averaged intensity measurements) to study these actin filaments and the intensity profiles on the graphs in the WT are completely distinct from the intensity profiles of the mutant (see Figure 2F, Figure 8, Figure 2—figure supplement 2, Figure 8—figure supplement 2).

Although the reviewer points out some patches of greater fluorescence under CD treatment, this can easily be explained due to compression effects, since the parasite replicates within a tight parasitophorous vacuole that also contains a dense food vacuole. It is therefore not surprising that the cytosolic signal is not evenly distributed.

In the case of CB-Halo the background is furthermore increased, since a soluble ligand has to be added for staining that diffuses into the cell.

While we agree that CD (and Jas treatment) alone is no definite proof for specificity, we also use conditional Act1 and Formin-2 knockout lines and demonstrate that the entire F-actin signal disappears upon removal of the gene proving specificity of CB for F-actin.

Finally, we would like to point out that CB is a reagent that has been validated and tested in several eukaryotes including its binding to F-actin in the apicomplexan parasite *T. gondii* (see also Periz et al., 2017). This commercial reagent is a well-accepted F-actin sensor and applied in many diverse eukaryotes. It appears to be especially well suited to visualise highly dynamic F-actin structures, such as nuclear F-actin (see Melak et al., 2017) or apicomplexan F-actin (Periz et al., 2017; Tosetti et al., 2019).

As a final biochemical proof, in the study ‘Reconstitution of the core of the malaria parasite glideosome with recombinant Plasmodium class XIV myosin A and Plasmodium actin’ by Bookwalter et al., (2017), the authors provide evidence for binding of Plasmodium actin filaments to actin-binding chromobodies.

In this respect, we would like to mention a recent, competing study, published in *eLife* that used our reagents (see acknowledgement of Tosetti et al., 2019), the CB in Toxoplasma and reaches similar conclusions regarding Formin-2 and F-actin dynamics in *Toxoplasma* (Tosetti et al., 2019).

Given the large background seen after Cytochalasin B as well as in the normal cells one needs some quantitative statements about comparative camera settings for image capture comparisons.

We performed cutting edge imaging and we did correct for background, used the same camera settings and exposure times. We are happy to provide the raw data for the reviewer’s information.

Figure 1C. The anti-actin antibody seems to give different staining and is deemed (subsection “Chromobodies label F-actin structures in *P. falciparum* asexual stages and in gametocytes”) to recognise both G and F actin. There is no biochemistry on the antibody – (blot of whole cell extracts, IP, etc.) so specificity not known. I don't feel confident about this conclusion when I compare the images.

The widely used anti-actin antibody has been previously published and characterised by the Baum group as suggested by this reviewer (Angrisano et al., 2012). We show that it does recognise the same structures and it is not surprising that slightly different staining patterns for F-actin can be obtained with different reagents. In this case, one also needs to consider that CB-Emerald binds actin prior to fixation and permeabilization in living cells, whereas the anti-actin antibody recognises the same structures after fixation.

We take the liberty of showing some images presented in Angrisano et al., which are fully consistent with the stain obtained with CB in our study:

Figure 4 Angrisano et al., 2012: Rings treated with or without Jas: Please compare to our Figure 1A,C;

Invasion of merozoite, Figure 6C of Angrisano et al., 2012: Please compare to our Figure 2.

Subsection “Chromobodies label F-actin structures in *P. falciparum* asexual stages and in gametocytes”. If chromobodies can bind to PfACT1 and actin2 (where is biochemical evidence?) then why conclude that the filaments could be built from both proteins? Why not of individual ones or a mix?

Since the actin-chromobodies used in our study can bind to actins from all eukaryotes tested so far and also to PfActin-1 (Bookwalter et al., 2017) there is no reason CB-EME will not bind actin-2. We would like to point out that based on expression (and what is known from other studies, see Hlisc et al., 2014), we would assume that PfAct-2 is detected in gametocyte. Again, please note that the stain we show in Figure 1 for gametocyte is almost identical to the F-actin stain presented in Hlisc et al., 2014 for gametocytes.

Subsection “Highly dynamic F-actin-rich filopodia-like structures extend outward from the periphery of growing parasites”. Structures are deemed to be highly dynamic and changing. This is very unscientific language given the data. What does it mean? Do the structures disappear and reappear? Do they move? Increase in size and decrease? Also, whatever they do can numbers not be given – time, number of events observed, number of types of events, etc.

The statements have been corrected. We will be happy to edit the text further, based on helpful comments from the reviewers.

Subsection “Chromobodies label F-actin structures in *P. falciparum* asexual stages and in gametocytes”. "passes through the whole cell" – unfortunate use of English.

We have corrected these statements.

Subsection “Chromobody labelled F-actin structures disappear upon disruption of PfACT1”. Figure 3F the authors speak about the close association of apicoplast with F actin. Actually, all one can see on this image is a large collection of dots. Why are there so many of different types if there should be only one apicoplast per cell?

The image depicts a schizont, meaning nucleus and apicoplast have already been divided. The apicoplast is then organised in a “tubular-like” network, before fission occurs, so that each merozoite obtains a single apicoplast (see vanDooren et al., 2005). Therefore, a staining pattern like the one shown in this study is fully expected.

Moreover, there is actually little coincidence by my eye of these dots. Surely the authors can quantify their statements by morphometric analysis. My conclusion is bolstered by the fact that there is very little yellow (overlapping green and red) on the merged image. Hence, their data don't justify their interpretation.

We have now provided quantifications for these.

Subsection “Chromobody labelled F-actin structures disappear upon disruption of PfACT1”. When disrupted they say that there is a consequential apicoplast segregation defect. I don't see this. In Figure 3F the apicoplast dots are smaller – why would this be if a segregation phenotype?Again, there are no numbers in the manuscript, no sizes, counts or morphometric analyses of juxtapositions.

The apicoplast segregates into individual apicoplasts, so that each merozoite will contain a single apicoplast at the end of schizogony. In the lower panel it can be seen that the apicoplasts clump together, resulting in a large “dot” that is not evenly distributed.

We have now provided detailed quantification for the phenotype in both organisms *Plasmodium* and *Toxoplasma*.

Subsection “DiCre-mediated conditional disruption of Formin-2 causes a defect in apicoplast segregation in *P. falciparum*”. The authors speak about collapsed or morphologically aberrant apicoplasts? What is a collapsed apicoplast? No correlative EM is used. They speak about collapsed/intermediate/normal apicoplasts without any definition. All one can see are a series of dots. There is nothing to provide a definition that a particular dot represents a particular "type" of apicoplast. Again, no numbers, counts etc. No statistics. What is a collapsed apicoplast and what does it look like at a sensible resolution – i.e. use electron microscopy.

We have now provided images to refer to what we define as normal / collapsed / intermediate apicoplasts. We had used a similar scoring strategy in our previous paper (Das et al., 2017)

One can continue with this type of analysis in further experiments.Actin filaments are known from previous work to be involved in apicoplast division and this paper seeks to implicate Formin in that process. This is a not unexpected factor to be involved. If the paper was to merit publication in such a journal it would need a stronger functional message based on more quantitative microscopy, some statistical analysis and some biochemistry. I am afraid this is a microscopy technique that seems to have overtaken the problem in a non-critical manner.

Again, we think this study needs to be seen in the context of the developments in the field, as appreciated by reviewer 1. It is for the first time possible to image F-actin in *Plasmodium* parasites and define its spatial localisation and dynamics over the asexual cycle of the parasite. A recent, competing study by Tosetti et al., 2019 (which used our reagents) comes to a very similar conclusions as our study. We also have defined this as research advance to Periz et al., 2017. Given that (A) *eLife* offers scoop protection and (B) this study is submitted as research advance (https://elifesciences.org/articles/03980). Therefore, we do think that *eLife* is the right platform for our manuscript.

To further strengthen this study, we have now provided functional data and shown that Formin-2 is responsible for maintaining a normal flow of F-actin in both *Plasmodium* and *Toxoplasma* using novel imaging tools.

[Editors' note: the author responses to the re-review follow.]

Reviewer #1:[…] The only larger concern I have is the cytokinesis phenotype in the Formin-2 KO (but this may be solved by simply changing the text to allow other options). Can the authors exclude that other phenotypes in later trophozoites or early schizonts contribute to the phenotype? Many different cellular defects could lead to the reduced number of nuclei in Figure 6A. In fact, this figure most likely indicates a general developmental delay rather than a specific phenotype. Do the 'slower' cells with less than 5 nuclei in Figure 6A still make it to egress? Or do they have other defects that make them arrest earlier? If they do not reach the late schizont, it may not only be a cell division phenotype. Was Figure 6E done with purified late stages or with all parasites on/off RAP?

We thank the reviewer for their positive comments and the critical reading of the manuscript. Data in Figures 6D and E are from experiments done with purified late stage parasites. The sentence in subsection “DiCre-mediated disruption of Formin-2 abrogates the actin network in *P. falciparum* schizonts” has been modified to clarify this. At this point we cannot rule out that additional defects in the late trophozoites cause a developmental delay. For example, it could be possible that actin plays a role in uptake of host cell material, as previously suggested. We are currently performing experiments to analyse this possibility in more detail and this will be subject to a later study.

We have changed the text to reflect this (see subsection “DiCre-mediated conditional disruption of Formin-2 affects daughter cell formation / cytokinesis in *P. falciparum”*).

The subsequent experiments (Figure 6B etc.) are then done with purified late stage parasites which consists of the subfraction that makes it to that stage (maybe those parasites where Formin-2 levels dropped more slowly after excision of the gene?). These experiments are important, as they show that the merozoites produced from this subfraction are still invasive and show that the Formin-1 dependent functions are not affected. The experiments are also consistent with the author's conclusion that this signifies a role of Formin-2 in 'coordinate cytokinesis'. However, it does not exclude that Formin-2 may have earlier function in schizogony. Unless all cells on RAP eventually make it to that stage, this may be just one of the phenotypes. This should be discussed.

We have now added explanatory sentences that a role for PfFRM2 earlier in schizogony cannot be ruled out (subsection “DiCre-mediated conditional disruption of Formin-2 affects daughter cell formation / cytokinesis in *P. falciparum”*). See also comment above.

Reviewer #2:[…] It would improve reading experience if the movies would be somehow fused into fewer total movies, if possible – e.g. Video 1, Video 2, Video 3 and Video 4 could be put back to back of each other into a single file.

We have considered this extensively, but we are of the opinion that stitching together movies will not necessarily improve the reading experience. Readers would have to scroll along a long movie to find the correct starting point for each parasite life-cycle stage – which would be highly inconvenient. In the online format of *eLife*, movies can be embedded at the correct position within the text, and that will anyway significantly improve the reading/viewing experience.

Can the structures visualized by the chromobody be bleached to investigate their turnover?

This experiment was already performed in the Periz et al., (2007), to which our current manuscript is a Research Advance. It was observed by FRAP that F-actin structures inside the parasite quickly recovered after photobleaching implying high turnover, but signal at extracellular F-actin tubules did not, indicating higher stability.

Does the bi-directional flow observed here compare with the recently reported actin flow in oocysts by the Heisenberg lab (2019)?

Shamipour et al., (2019) recently showed in zebrafish oocytes that a wave of actin polymerization is responsible for segregation of yolk granules (down towards the vegetal pole) and ooplasm (up towards the animal pole). Additionally, actin comets formed on the yolk granules provide further force to push these granules down and actin flows pull ooplasm upwards. Parallels can be drawn from this study to our own. It is conceivable that actin polymerization and hence flow derived forces set up by Formins aid in particle segregation, specifically as we observe for segregation of the apicoplast, and as has already been highlighted in our model in Figure 9. We have now added the Shamipour et al., reference to our Discussion section.

How does this fit to the data recently published in eLife, where actin flows only from front of T. gondii to their back?

Most recently, Tosetti and co-workers published their investigation of the three actin nucleation factors in Toxoplasma (Tosetti et al., 2019). This study shows that TgFormin1 does not majorly contribute to intracellular actin dynamics. Instead, TgFormin1 appears critical for motility, invasion and egress of extracellular parasites. Results presented by Tosetti and colleagues regarding TgFormin-2 are in good agreement with data presented here.

TgFormin1-mediated actin nucleation was described to be critical for maintaining actin flow in extracellular parasites (Tosetti et al., 2019). According to live microscopy studies of moving and non-moving extracellular parasites, actin appears to accumulate at the basal end of the parasite, indicating actin flow from the apical to the basal pole. Tosetti and co-workers suggest that actin translocation to the basal pole happens along the parasite periphery (Tosetti et al., 2019). Our study provides experimental evidence that the parasite periphery represents one site of increased actin abundance during intracellular growth. Actin flow appears bi-directional along the periphery in intracellular parasites and is fuelled by TgFormin-2-mediated actin polymerisation.

In conclusion, it appears that the peripheral actin flow depends on TgFormin1 in extracellular parasites while depending strongly on TgFormin-2 in intracellular parasites. The one-directional actin flow to the basal end in extracellular parasites was proposed to mediate parasite motility (Tosetti et al., 2019). In intracellular parasites, when actin does not promote parasite movement, bi-directional actin flow could be involved in a vast variety of cellular processes. Research elucidating these processes is on-going in our laboratory. At this point of time, we are hypothesising that bi-directional actin flow might, amongst other things, allow for particle trafficking within replicating parasites.

We have already discussed this in various parts of the manuscript. See Results section and Discussion section.

Author response image 1 summarizing the same has been presented for the reviewer’s reference:

**Author response image 1. respfig1:** Actin flow in Toxoplasma.

Reviewer #3:[…] 1) The authors frequently use the term "dynamics" when describing actin-based structures. Based on their data, I would agree that many of the structures they view are dynamic in the sense that are mobile (e.g. subsection “F-actin in gametocytes”). However, "dynamic" with regards to actin also often refers to the degree to which the actin filaments undergo turnover (rounds of polymerization and depolymerization). The authors should review their text and edit where it needs to be made clear which type of dynamics they are discussing. Related to this, the authors claim to show that "islands of F-actin are stabilized upon jasplakinolide treatment (+JAS) (Figure 1C)", but this is documented only by a still image with large fluorescent bodies. The authors cite Video 3, but there is no +JAS movie on that video, only control and +CytoD. I agree that the authors could interpret their result as being consistent with stabilization of F-actin, but without showing that there is reduced turnover of actin in these structures, they should hedge on their wording some.

We thank the reviewer for their comments. We have now included a +JAS panel in Video 3 to show that actin dynamics are indeed slowed down and the “islands” of F-actin are indeed stabilized.

2) I also strongly recommend the authors supplement some of their figures/figure legends to help the reader better interpret the data. In particular:For Figure 2Bii, the legend explains the significance of the bottom panels, but not the top four panels. Are they different time points?

This has been clarified and the fact that the panels are different time points has been elaborated in the figure and in the legend.

For Figure 3G, the figure legend should provide some more information to help the reader understand the model. That is, "syn", "loxPint", and "WR" should be defined. It is a very elegant construct but requires a lot of flipping back and forth between the legend, text, and methods to deconstruct what is going on.

Explanatory sentences in the legend and definitions of all labels have been now provided.

For Figure 3H, it currently gives the impression that Integrant 1 and Integrant 2 are being analyzed, when in fact it is transformant 1 and transformant 2 that are being analyzed, using a PCR reaction that probes for integration or the endogenous locus. I would suggest clarifying the labels, such as replacing "Integrant" and "Endogen." with "PCR(Int)" and "PCR(End)" (with clarifications in the legend), and add the label "Transformant:" to the left of "1 2 1 2" above the gels.

These changes have been made.

For Figure 4C, the x-axis is confusing to me. Perhaps this is because I do not study *P. falciparum*, but that will be true of a large number of readers.

Growth over three 48 hour replication cycles have been measured. Various time points in each cycle has been depicted on the x axis. The unit hours (h) was partially hidden behind a panel and is now clearly visible. Also, the fact that first, second and third represent replication cycles has been clearly stated in the figure and the legend.

For Figure 4D, what are the units for the y-axis? Is it an arbitrary value? A ratio of HA signal to enolase?

Protein levels have been measured by quantification of intensity and normalized to enolase (now stated in the legend). The y axis is in arbitrary units and has now been clarified in the legend and figure.

For Figure 7—figure supplement 1, reproduce the map shown in Figure 7A to help the reader to interpret the PCR results without needing to flip between two figures.

Done.

For Videos 9 – 11, multiple strains are shown simultaneously, but I could not observe any labels that identified which panel in a video corresponded to which strain. This was true when I viewed the movies directly from the website, and when I downloaded the mp4's.

We now make sure that each panel is labelled with the name of the strain and the Movie number, which corresponds to strains in the figures. WT and mutant have also been separated with solid lines.

3) Related to the phylogenetic analysis, Table T2, which lists the genomes that were searched for formins for the phylogenetic analysis, appears to be absent. Also, Figure 3B presents a single tree, but the Materials and methods section refers to "trees" (plural) that were constructed from "this alignment (or subsets of it)". The authors should clarify this. It would also be appropriate to provide the aligned sequences as a supplementary text file, if possible.

Necessary changes have been to clarify that a single tree was generated. We apologise for the oversight of not providing Table T2. We have added this to the submission.